# RETHINKING MULTI-OMICS LLMS FROM THE PERSPECTIVE OF OMICS-ENCODING

## ABSTRACT

Understanding living systems requires interpretable models to elucidate how multi-omics data coordinate transcription and translation across spatiotemporal scales. Inspired by large language models (LLMs), biological foundation models pretrained the omics sequences have shown exciting performance. However, these biological models lack interpretability and transparency in explaining the results. Motivated by advances in cross-modal alignment from vision–language models (VLMs), it is naturally to integrate multi-omics data and nature language into one system: multi-omics large language model (MOLLM), a LLM-based model can understand multi-omics data. To understand the trends, challenges, and limitations of MOLLMs, we provide a comprehensive empirical study on MOLLMs. We systematically review recent progress on MOLLMs based on their omics-encoding design and benchmark the performance gap between MOLLMs with omics-specific models. The extensive experiments show that the proposed multi-omics-encoding design outperforms existing MOLLMs by a large margin and shows promise for narrowing the performance gap against specialist biological models. Code is available at https://anonymous.4open.science/r/mollm.

## 1 INTRODUCTION

Multi-omics data operate under the central dogma (Crick, 1970) through hierarchical and coupled mechanisms to coordinate transcription and translation. This coordination shapes phenotypes across spatiotemporal scales and plays a key role in trait prediction, disease mechanisms and drug design (Elzawahry et al., 2014). Therefore, deepening the multi-omics understanding is essential for explaining the complexity of biological processes and for advancing technologies in agriculture, medicine, and pharmacy(Śledź & Caflisch, 2018; Le, 2020; Ahmar et al., 2020).

In recent years, large language models (LLMs, *i.e.* generalist models) such as GPT-4 (Achiam et al., 2023) have excelled at natural language tasks. With modest instruction tuning, they often achieve usable performance in specialized domains (Wu et al., 2024; Yang et al., 2025b). Notably, multi-omics data and natural language share similar fundamental structural properties, namely discrete symbolic alphabets, compositional grammars, and hierarchical contextual dependencies (Dotan et al., 2024; Sanabria et al., 2024). Thus, biological foundation models are inspired by LLM techniques. Pre-trained in large-scale omics sequence data, they capture domain-critical signals such as cis-regulatory motifs (Zhou et al., 2023) and long-range dependencies (Dalla-Torre et al., 2025). Consequently, as illustrated in Fig. 1 (a), the biological foundation models are used to encode raw omics sequences into embeddings for specific downstream tasks (Sanabria et al., 2024; Yuan et al., 2025). In this work, we denote these models as omics-specific models (*i.e.* specialist models). These approaches typically produce direct task predictions without natural language responses or explainability. And they always require redesigning or retraining when new tasks arise, which limits generalization. A separate line of work explores tool-calling agent pipelines that orchestrate external bioinformatics tools, such as biological foundation models. Although built on LLMs, agent-based methods lack interpretability and transparency in explaining the results (Shen, 2024).

Notably, the success of coupling LLMs with vision encoders, exemplified by vision-language models (VLMs) (Achiam et al., 2023; Dai et al., 2023; Liu et al., 2023), shows that cross-modal alignment could outperform unimodal approaches. Grounding visual inputs in natural language unifies diverse downstream tasks. And enables VLMs to inherit interpretability and instruction-following

capabilities from LLMs. Building on this paradigm, multi-omics large language models (MOLLMs) that integrate multi-omics sequences and human language have naturally emerged. The similarities between multi-omics data and natural language make them naturally amenable to integration into a common representational space (Nam et al., 2024). Based on the observations from VLMs, it is conjectured that multimodal integration can improve the biological task performance and interpretability by leveraging the instruction-following capabilities and linguistic knowledge acquired through large-scale pretraining on natural language.

To better understand the challenges of the multimodal integration mentioned above, we first review the recent progress of MOLLMs with a focus on DNA, RNA, and protein, which are key modalities in omics study (Karczewski & Snyder, 2018). In this work, we categorize existing MOLLMs into three types in terms of omics encoding: non-omics encoding (NOE), which omits biological encoders; single-omics encoding (SOE), which uses a shared biological encoder; and multi-omics encoding (MOE), which employs multiple biological encoders (Fig. 1). For NOE, recent approaches (He et al., 2025; Xia et al., 2025) fine-tune LLMs by treating multi-omics sequences as text data. However, they overlook structural and biophysical priors intrinsic to biological data. As a result, these methods fall short compared with omics-specific models. Meanwhile, several studies (Wang et al., 2024; Fallahpour et al., 2025; Lv et al., 2025) adopt SOE design and have shown strong performance against NOE methods. They use a single-omics biological foundation model to encode omics sequences into embeddings, which are then fed into an LLM. In this way, biological priors are combined with the instruction-following capability of LLMs. Commonly, SOE methods focus on single omics tasks and keep the language component frozen or only lightly tuned, which limits generality across omics. An exception is ChatNT (de Almeida et al., 2025), which projects RNA and protein sequences into a single DNA-encoder. However, this approach sacrifices cross-modal information and prevents end-to-end representational alignment. There are limited studies on MOE. In this work, we design the first prototype MOE model for DNA, RNA, and protein.

Despite above progress, there is still a non-trival performance gap between MOLLMs and omics-specific models on biological tasks. To systematically understand the current performance gap and the capability limit of existing MOLLMs, it is necessary to quantitatively benchmark these models in a comprehensive fashion. Following ChatMultiOmics (He et al., 2025), we select 9 common biological tasks across single omics and multi-omics. In addition to the empirical understandings of MOLLMs, the extensive experiments show that MOE outperforms general LLMs, NOE models and SOE models by a large margin and shows promise for narrowing the performance gap against omics-specific models. The empirical findings also suggest insights to future MOLLM design and generalist-specialist learning paradigm (Shi et al., 2023).

Our contributions are summarized as follows:

- We present a comprehensive empirical study to understand the trends, challenges, and limitations of MOLLMs. We systematically review recent progress on MOLLMs based on their omics-encoding design and benchmark the performance gap between MOLLMs with omics-specific models.

- We conduct extensive experiments fo facilitate the research on MOLLMs. The empirical results show that the proposed MOE design achieves the most robust performance in omics sequence understanding and has the potential to narrow the performance gap with omics-specific models.

## 2 RELATED WORK

Recent efforts to create MOLLMs have explored several distinct architectural paradigms. One straightforward approach fine-tunes LLMs by treating multi-omics sequences purely as text, without a modular encoder-LLM separation. Representative models like ChatMultiOmics (He et al., 2025), Intern-S1 (Bai et al., 2025a) and NatureLM (Xia et al., 2025) exemplify this direction, illustrating the feasibility of processing DNA, RNA, and protein tasks within a unified textual framework by undergoing continued pre-training on hundreds of billions of scientific sequence tokens or fine-tuning on large-scale instruction–answer pairs. However, this method can overlook the rich structural and biophysical priors intrinsic to biological data.

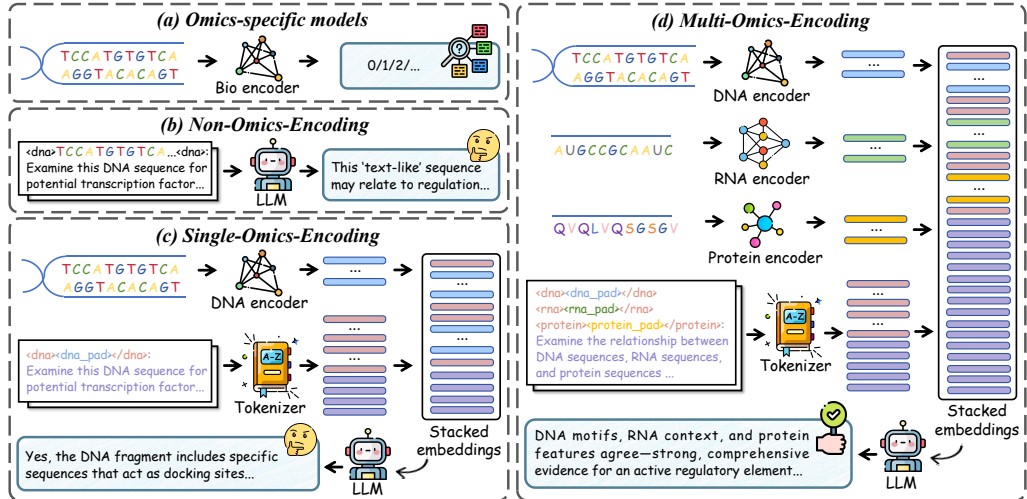

Figure 1: Comparison of four paradigms for omics sequence understanding. (a) Omics-specific models: biological encoders (e.g., DNA, RNA, or protein) trained or fine-tuned for specific tasks without LLMs; (b) Non-Omics-Encoding: omics sequences treated as plain text for instruction tuned LLMs to generate natural language outputs; (c) Single-Omics-Encoding: a single biological foundation model (e.g., DNA, RNA, or protein) encodes omics sequences into embeddings that are fed into an instruction tuned LLM to generate natural language outputs, illustrated with DNA; (d) Multi-Omics-Encoding: integration of multiple biological encoders (e.g., DNA, RNA, and protein) with instruction-tuned LLMs to support multi-omics understanding and natural language outputs.

To better incorporate these priors, a more prevalent strategy is to align a specialized biological encoder with an LLM (Nam et al., 2024; Fallahpour et al., 2025; Lv et al., 2025). Large-scale pre-training has produced powerful foundation models ideal for this role, such as Evolutionary Scale Modeling (ESM) for proteins (Lin et al., 2022) and Nucleotide Transformer (NT) for DNA (Dalla-Torre et al., 2025). Consequently, many studies integrate such encoders to combine domain-specific knowledge with the instruction-following abilities of LLMs. For instance, BioReason embeds DNA encoders into LLMs to enable multi-step biological reasoning (Fallahpour et al., 2025). In proteomics, ProtChatGPT (Wang et al., 2024) builds an interactive protein analysis system by connecting a protein encoder to an LLM via a protein-language pretraining transformer and a projection adapter. Generative approaches like ProtT3 (Liu et al., 2024) connect a protein encoder to a text decoder to enable protein-to-text generation. Meanwhile, domain-focused methods include MutaPLM (Luo et al., 2024), which is trained to generate natural language explanations for mutation effects. A notable variant within this paradigm is ChatNT (de Almeida et al., 2025), which is based on a DNA-centric encoder (NT-v2) and maps RNA and protein sequences into DNA-like representations to enable cross-omics reasoning. Building on these pioneering efforts, such models show the potential of aligning biological priors with LLMs. However, they are often limited by their reliance on the perspective of single-omics encoding, which restricts cross-omics data fusion. In parallel, the emergence of large-scale datasets like Biology-Instructions (He et al., 2025) offers the possibility of training and rigorously benchmarking a more comprehensive multi-omics-encoding paradigm. This convergence of prior limitations and new resources sets the stage for our work to evaluate these three paradigms and quantify the performance gaps between them and with domain-specialist models.

## 3 METHOD

The overall framework is shown in Fig. 1 (d). This multi-omics-encoding model accepts queries that contain one or more biological sequences together with a textual prompt. It supports three sequence types: DNA, RNA, and protein. Let, $\mathcal{K} = \{\mathrm{D}, \mathrm{R}, \mathrm{P}\}$ denote the set of sequence modalities (DNA, RNA, protein). For $k \in \mathcal{K}$, define the input sequence collection as $S_k = (s_{k,1}, s_{k,2}, \ldots, s_{k,\ell_k})$, where $\ell_k \in \mathbb{N}$ is the length of sequences provided for modality $k$. And let $Q$ denote the textual query.

| Group | Model | TF-M MCC | PD300 MCC | CPD MCC | ncRNA ACC | Modif AUC | EC Fmax | Sol ACC | AAN MCC | RPI MCC | Avg. |
|---|---|---|---|---|---|---|---|---|---|---|---|
| **General LLMs** | DeepSeek-R1 | 1.33 | −0.99 | −0.89 | 7.49 | 50.97 | 5.68 | 53.93 | 2.48 | −1.44 | 13.17 |
| | GLM4-9B-Chat | 0.00 | −0.25 | −2.53 | 8.23 | 50.05 | 0.91 | 50.72 | 1.32 | 0.13 | 12.06 |
| | GPT-4o | −1.38 | 8.67 | −0.84 | 5.60 | 50.47 | 5.89 | 51.67 | −3.29 | 1.17 | 13.11 |
| | Qwen3-1.7B | 1.08 | −1.84 | −4.24 | 6.87 | 51.15 | 1.01 | 50.42 | −2.79 | −1.18 | 11.16 |
| | Qwen3-4B | −0.01 | −0.59 | −5.29 | 6.68 | 50.24 | 1.82 | 50.27 | −2.14 | 1.96 | 11.44 |
| | Qwen3-8B | −0.03 | −3.57 | −3.64 | 7.25 | 50.15 | 1.09 | 49.88 | 0.37 | −1.74 | 11.08 |
| | Qwen3-32B | −0.9 | −1.77 | 0.62 | 7.40 | 51.66 | 1.41 | 51.62 | 3.33 | −0.07 | 12.58 |
| **MOLLMs** | **Non-Omics-Encoding** | | | | | | | | | | |
| | ChatMultiOmics | 32.21 | 56.13 | 44.19 | 63.09 | 59.06 | 19.79 | 63.02 | 1.06 | 74.26 | 45.87 |
| | Intern-S1 | 1.81 | −1.14 | −1.02 | 7.49 | 50.69 | 8.56 | 48.75 | −5.24 | 1.47 | 12.37 |
| | NatureLM* | 3.75 | 25.94 | 14.71 | 0.00 | 51.54 | 11.81 | 48.93 | 0.68 | 5.33 | 18.08 |
| | **Single-Omics-Encoding** | | | | | | | | | | |
| | BioReason | 34.39 | 53.34 | 43.84 | – | – | – | – | – | – | 43.86 |
| | ChatNT* | 2.11 | 42.90 | 14.83 | 0.00 | 51.93 | – | – | – | – | 22.35 |
| | ProLLaMA* | – | – | – | – | – | 1.85 | 49.53 | 0.00 | – | 17.13 |
| | **Multi-Omics-Encoding** | | | | | | | | | | |
| | Ours-1.7B | 39.55 | 62.69 | 49.80 | 78.50 | **67.85** | 49.52 | **67.50** | 32.85 | 70.70 | 57.66 |
| | Ours-4B | 39.49 | 62.03 | 50.56 | 83.75 | 66.68 | 55.45 | 66.69 | 34.76 | 70.70 | 58.90 |
| | Ours-8B | **40.20** | **62.85** | **50.97** | **84.74** | 67.51 | **66.51** | 67.39 | **40.52** | 71.20 | **61.32** |
| **OSMs** | DNABERT2 | 71.21 | 83.81 | 71.07 | – | – | – | – | – | – | – |
| | GCN | – | – | – | 85.73 | – | – | – | – | – | – |
| | MultiRM | – | – | – | – | 84.00 | – | – | – | – | – |
| | SaProt-GearNet | – | – | – | – | – | 88.90 | – | – | – | – |
| | DeepSol | – | – | – | – | – | – | 77.00 | – | – | – |
| | DeepAAI | – | – | – | – | – | – | – | 54.90 | – | – |
| | ncRPI-LGAT | – | – | – | – | – | – | – | – | 93.20 | – |

Table 1: Overall performance comparison across nine biological omics tasks for different model categories. All MOE results are arithmetic means over three runs. "–" denotes a task not supported by the model. Results under Omics-Specific Models (OSMs) are shown in gray to indicate the hypothetical task upper bound in this work. To highlight strong performers, in each column (excluding OSMs) the top two entries are emphasized: **bold** marks the best and underline marks the second best. Results marked with * are obtained from publicly released checkpoints. Standard deviations, task descriptions, and metric definitions are provided in the Appx. C.3.

For each biological sequence $S_k$, respectively, we use the modality-specific biological encoder $\mathrm{Enc}_k$ to obtain contextualized representations. We first apply the tokenizer $T_k$ associated with $\mathrm{Enc}_k$ to segment the sequence into tokens. A single token may correspond to one or more nucleotides for DNA and RNA, whereas each token corresponds to one amino acid for proteins. The encoder $\mathrm{Enc}_k$ then maps the token sequence to token-level embeddings $E_k \in \mathbb{R}^{L_k \times d_k}$ that capture rich biological information, where $L$ denotes the token sequence length and $d$ denotes the dimensionality of the token-level embeddings.. These embeddings $E_k$ are subsequently provided to the LLM as inputs for training and inference.

The LLM serves as the primary reasoning and text generation module and must integrate the token-level embeddings $E_k$ produced by the biological encoders. Because embedding dimensionalities differ across modalities, we introduce a modality-specific linear projection matrices $W_k$ to map $E_k$ into the LLM embedding space. Given $E_k$, the projected representations $Z_k \in \mathbb{R}^{L_k \times d_{\mathrm{LLM}}}$ are computed as:

$$Z_k = E_k W_k, W_k \in \mathbb{R}^{d_k \times d_{\mathrm{LLM}}} \tag{1}$$

Here $d_{\mathrm{LLM}}$ denotes the token-embedding dimension of the language model.

Finally, we construct the LLM input $X$ by concatenating the embedding of the textual query $Q$ with the projected embeddings $Z_k$ corresponding to each biological sequence. The concatenation order preserves the original placement of each sequence within the prompt; sequences may appear at arbitrary positions in $Q$, and the embedding blocks are interleaved accordingly.

# 4 EMPIRICAL STUDY

## 4.1 EXPERIMENTAL SETTINGS

**Datasets.** To train a MOE capable of understanding DNA, RNA, and protein sequences, we use the Biology-Instructions dataset (He et al., 2025), from which we select approximately 700K question and answer pairs. The data cover three omics modalities: DNA, RNA, and protein, and span nine tasks, including binary classification, multi-class classification, and multi-label classification. We then systematically evaluate the MOLLM and a set of baseline methods on these nine tasks. Specifically, the nine tasks are organized into four categories by input format. (1) Single-DNA inputs cover transcription factor binding site detection in mouse (TF-M), promoter detection (PD300), and core promoter detection (CPD); (2) single-RNA inputs cover non-coding RNA function classification (ncRNA) and RNA modification prediction (Modif); (3) single-protein inputs cover Enzyme Commission number prediction (EC) and protein solubility prediction (Sol); and (4) multi-sequence inputs cover antibody-antigen neutralization (AAN), which takes two protein sequences, and RNA-protein interaction prediction (RPI), which takes one RNA sequence and one protein sequence. Dataset statistics and task descriptions are provided in the Appx. B.

**Evaluation Metrics.** In line with prior work (He et al., 2025), we evaluate TF-M, PD300, CPD, AAN, and RPI using the Matthews correlation coefficient (MCC); ncRNA and Sol using accuracy (ACC); Modif using area under the ROC curve (AUC); and EC using Fmax. Definitions and computational details are provided in the Appx. B.3.

**Implementation Details.** We use Qwen3 (Yang et al., 2025a) models with 1.7B, 4B, and 8B parameters as the language backbone, and adopt NT-500M and ESM2-650M as the omics encoders. Following the LLaVA training paradigm (Liu et al., 2023), we freeze the omics encoders and update only the language model parameters and the modality-specific projection matrices. All experiments are conducted on eight A100 GPUs. Full training configurations and implementation details are provided in the Appx. C.1 .

**Baselines.** To systematically benchmark MOLLMs on multi-omics biological sequence understanding and quantify their performance gap against omics-specific models, we adopt a unified evaluation protocol and organize baselines into three categories: **(1)** omics-specific models, domain-specialist models tailored to specific biological tasks that do not rely on natural language pretraining; **(2)** general large language models (General LLMs); and **(3)** multi-omics large language models (MOLLMs), which can process both omics sequences and natural language. For each task, we treat the omics-specific models as the performance upper bound.

Across nine benchmark tasks, we select omics-specific models that are widely recognized as leading: DNABERT2 (Zhou et al., 2023) for TF-M, PD300, and CPD; GCN (Rossi et al., 2019) for ncRNA; MultiRM (Song et al., 2021b) for Modif; SaProt-GearNet (Su et al., 2024) and DeepSol (Khurana et al., 2018) for EC and Sol, respectively; and DeepAAI (Zhang et al., 2022b) and ncRPI-LGAT (Han & Zhang, 2023b) for AAN and RPI, respectively. For general LLMs, we adopt representative baselines including DeepSeek-R1 (DeepSeek-AI, 2025), GLM4-9B-Chat (GLM et al., 2024), GPT-4o (Hurst et al., 2024), and Qwen3-1.7B, Qwen3-4B, Qwen3-8B, and Qwen3-32B (Yang et al., 2025a).

For MOLLMs, we adopt a finer-grained taxonomy from the perspective of omics-encoding: NOE, SOE, and MOE. Specifically, we categorize ChatMultiOmics (He et al., 2025), Intern-S1 (Bai et al., 2025a), and NatureLM (Xia et al., 2025) as NOE, and BioReason (Fallahpour et al., 2025), ChatNT (de Almeida et al., 2025), and ProLLaMA (Lv et al., 2025) as SOE. All models are evaluated under matched experimental settings using task-aligned metrics. The complete list of models, detailed descriptions, versions, and configurations is provided in the Appx. C.2.

## 4.2 PERFORMANCE COMPARISON WITH STATE OF THE ART

As shown in Tab. 1, we systematically compare overall performance on multi-omics sequence tasks under a unified evaluation protocol, covering omics-specific models, general LLMs, and multiple MOLLM paradigms, we provide all the additional results in Appx. C.3. First, general LLMs, whether proprietary (e.g., GPT-4o) or open source (e.g., DeepSeek-R1), exhibit limited performance on multi-omics sequence tasks, indicating that generic linguistic capability alone is insuf-

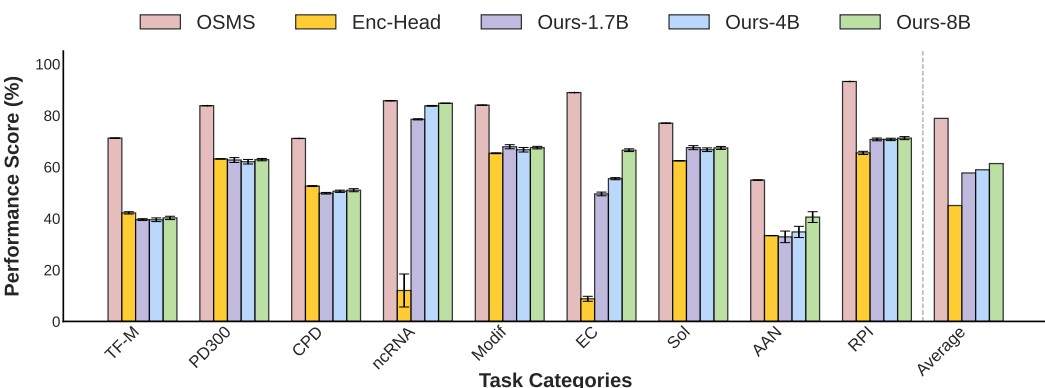

Figure 2: Performance comparison between the MOE and two baseline families, namely (i) high-performing omics-specific models (OSMs) and (ii) methods that use only the corresponding omics encoder with a classification head (Enc-Head). Overall, MOE achieves improvements over Enc-Head on several tasks and narrows the performance gap with OSMs.

ficient for biological sequence understanding. Second, among MOLLM designs, approaches that employ omics encoders consistently outperform those that do not. Using ChatMultiOmics as a point of comparison, which introduces an additional pretraining stage beyond supervised finetuning to inject biological sequence knowledge, does not adopt omics encoders, and uses an 8B parameter size. MOE approach leads on eight of nine tasks and is comparable on the remaining one; even with supervised finetuning only, our 1.7B variant surpasses ChatMultiOmics. Furthermore, relative to single omics encoder approaches, MOE not only achieves superior performance on most metrics but also covers a broader range of task types and modality combinations. For example, the RPI task involves both RNA and protein sequences, which single omics encoders cannot directly handle. Taken together, under the current evaluation setup, MOE outperforms other MOLLM variants overall and represents a promising direction for future research.

## 4.3 COMPARISON OF MOLLMS WITH OMICS-SPECIFIC MODELS

Although many MOLLMs can handle multiple modalities and a variety of tasks within a unified framework and achieve substantial gains over general natural language LLM baselines, this alone does not demonstrate genuine fusion of generalist and specialist capabilities, namely equipping a general-purpose model to solve domain-specialized tasks while retaining generality. In practice, carefully engineered omics-specific models remain superior on most single-task settings. To assess this gap systematically, we compiled omics-specific models recognized as leading on their respective tasks across nine benchmarks and compared them with MOLLMs (see Tab. 1 and Fig. 2). The results show that MOLLMs lag behind task-specific approaches on most single tasks. An exception emerges on the more challenging ncRNA multi-class classification task, where the MOE approaches the performance of leading domain models, suggesting that the paradigm can reach specialist-level accuracy while preserving natural language output and interaction capabilities. t is worth noting that specialist models require bespoke design and training for each task, whereas MOE shows greater generality and scalability by covering multiple tasks with a single model.

Because the MOE approach incorporates the omics encoders NT and ESM, we further evaluated their representational capacity on the target tasks. After adding a linear classification head to NT and ESM and fine-tuning, performance improved on more than half of the tasks (see Fig. 2). This indicates that coupling a large language model enhances biological sequence understanding and facilitates task adaptation. It also implies that omics-specific models usually need additional task-aware architectural and training design to achieve strong results, rather than simply attaching a linear head, which runs counter to the goal of generality. In addition, EVO2, a leading backbone model in genomics, was evaluated on our DNA task, with results reported in the Fig. 3 (a). EVO2-7B attains higher scores than Ours-4B overall; however, on the CPD task our model reaches a comparable level (50.56 vs. 52.70), indicating that MOE has strong potential.

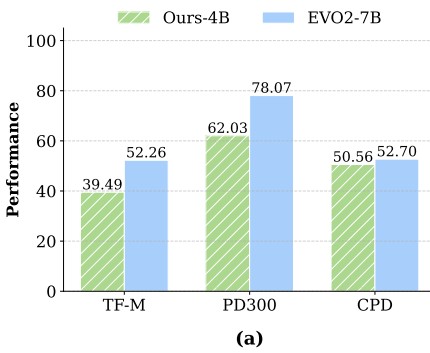 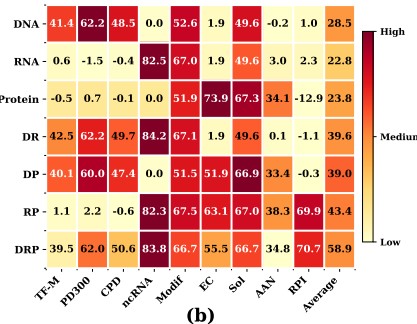

Figure 3: (a) Performance comparison between Ours-4B and EVO2-7B on DNA tasks. Results are reported on TF-M, PD300, and CPD, with higher values indicating better performance. EVO2-7B attains higher scores on all three tasks; (b) Heatmap of MOE performance across tasks (x axis) and omics combinations (y axis). The Avg. column reports the mean score for each combination. Omics combinations denote the training data: single omics DNA, RNA, or Protein; pairwise combinations DNA and RNA (DR), DNA and Protein (DP), and RNA and Proteins (RP); and DNA, RNA, and Protein (DRP), which uses all available omics data.

Overall, the MOE approach is narrowing the gap to omics-specific models while retaining high generality: a single model can address multiple tasks. These properties indicate strong promise for MOE within the MOLLM paradigm.

## 4.4 ANALYSIS OF CROSS-OMICS TRANSFER

To examine cross-omics knowledge transfer, we analyze a 4B setting where the model is trained on single-omics datasets (DNA, RNA, Protein) and on each pairwise omics combination; results are shown in Fig. 3 (b). We find that, for DNA-related tasks, adding RNA yields consistent gains, whereas adding Protein alone provides no clear improvement. On CPD, using all three omics performs best. For RNA-related tasks, adding DNA likewise improves performance. For protein-specific tasks, training on Protein alone typically achieves the best results. For multi-sequence or cross-omics interaction tasks, the RNA–Protein pairing strengthens AAN, and RPI obtains its best results under joint training with all three omics. We find that modality gains are task-dependent and reflect complementarity. Consequently, adding more modalities is not universally beneficial. In DNA and RNA tasks, the two modalities typically complement each other. In protein-specific tasks, introducing non-target modalities can dilute key signals, yielding limited benefits or even mild negative transfer. For tasks that require cross-omics sequence integration, such as RPI, multimodal joint training is more advantageous.

## 4.5 ANALYSIS OF MULTI-TASK EFFECTS

To evaluate whether a larger variety of training tasks improves performance for MOLLMs with a MOE design, in other words whether multitask training enhances shared representations and promotes cross task transfer, we conduct a 4B study under identical training settings. We train separate models on each of the nine tasks and another model on the union of all nine tasks, and we compare their performance; results are shown in Fig. 4. Overall, multi-task training yields substantial gains on most tasks, with the largest benefits for tasks that rely on cross modal or cross sequence information. For example, ncRNA accuracy increases from 44 to 84, and AAN MCC rises from near zero to 35; PD300, CPD, Sol, and RPI also improve to varying degrees. In contrast, EC performs better under single task training, which suggests that supervision from other tasks can introduce negative transfer for protein specific functional prediction. Taken together, these results show that MOLLMs with a MOE design benefit from diverse task training, though the gains are task dependent.

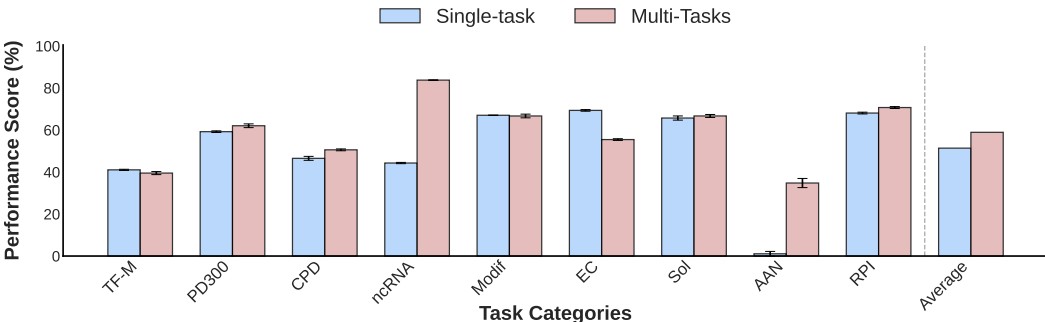

Figure 4: Performance comparison of the MOE under single task training (Single-Task) versus joint training on nine tasks (Multi-Tasks). With the exception of a few tasks such as EC, Multi-Task yields better performance than Single-Task.

| Scale | TF-M | PD300 | CPD | NcRNA | Modif | EC | Sol | AAN | RPI | Average |
|---|---|---|---|---|---|---|---|---|---|---|
| | MCC | MCC | MCC | ACC | AUC | Fmax | ACC | MCC | MCC | |
| 10% | 28.31 | 49.08 | 36.75 | 9.59 | 55.55 | 3.19 | 60.32 | 0.95 | 1.22 | 27.22 |
| 20% | 31.78 | 54.06 | 39.15 | 22.34 | 60.93 | 5.46 | 61.45 | 20.94 | 59.05 | 39.46 |
| 30% | 37.65 | 58.96 | 44.19 | 42.59 | 65.48 | 6.85 | 65.10 | 22.63 | 65.09 | 45.39 |
| 40% | 38.40 | 59.64 | 47.11 | 56.38 | 66.22 | 24.91 | 66.14 | 28.44 | 66.72 | 50.44 |
| 50% | **40.25** | 60.30 | 45.84 | 71.26 | 66.04 | 40.92 | 66.05 | 30.18 | 67.00 | 54.20 |
| 100% | 39.49 | **62.03** | **50.56** | **83.75** | **66.68** | **55.45** | **66.69** | **34.76** | **70.70** | **58.90** |

Table 2: Investigating the effect of training data scale on MOE performance. For each task, we uniformly sample 10%, 20%, 30%, 40%, and 50% of the training set and compare against using 100% of the data. Reported values are arithmetic means over three independent runs. In each column, including the Average column, the top two entries are highlighted: **bold** indicates the best and underline indicates the second best. Most tasks reach peak performance at 100% data. Detailed experimental settings and results are provided in the Appx. C.4.

## 4.6 Ablation study

In general LLM research, model performance typically improves with larger parameter counts and more training data (Achiam et al., 2023; Isik et al., 2024), and in VLMs, higher image resolution also yields better results (Bai et al., 2025b). Whether these conclusions hold true for MOLLMs with MOE remains an empirical question that requires further investigation. In this section, we present an ablation study across nine tasks to evaluate the effects of parameter scaling, data scaling, and omics sequence length within the MOE setting.

**Parameter Scaling.** We evaluate the performance of MOE with language backbones of 1.7B, 4B, and 8B (see Tab 1). The results observed in the table clearly indicate that, by task type, ncRNA, EC, AAN, and RPI, which involve multiclass, multilabel, and multi-sequence settings, are particularly sensitive to model scale. In contrast, binary classification tasks such as TF-M, PD300, and CPD are less sensitive to the parameter scale, which may be influenced by the inherent difficulty and complexity of the tasks. Overall, it can be observed that for models with parameter sizes up to 8B, the MOE follows the scaling law with respect to parameter scale, showing consistent performance improvements as the model size increases.

**Data Scaling.** Under identical training settings we fine tune a 4B MOE using task-stratified random samples comprising 10%, 20%, 30%, 40%, 50%, and the full 100% of the training set (see Table 2). Overall, with the exception of TF-M, the other eight tasks achieve their best results with the full dataset. Task responses to data scale vary across different tasks: several tasks saturate between 30% and 50% of the data, and as a result, additional data provides only limited marginal gains, as seen in tasks like PD300, Modif, and Sol. This suggests that task difficulty plays a significant role in data demand, where relatively simpler or more redundant tasks show less sensitivity to data size. At the same time, certain tasks exhibit clear threshold effects, indicating that performance improves

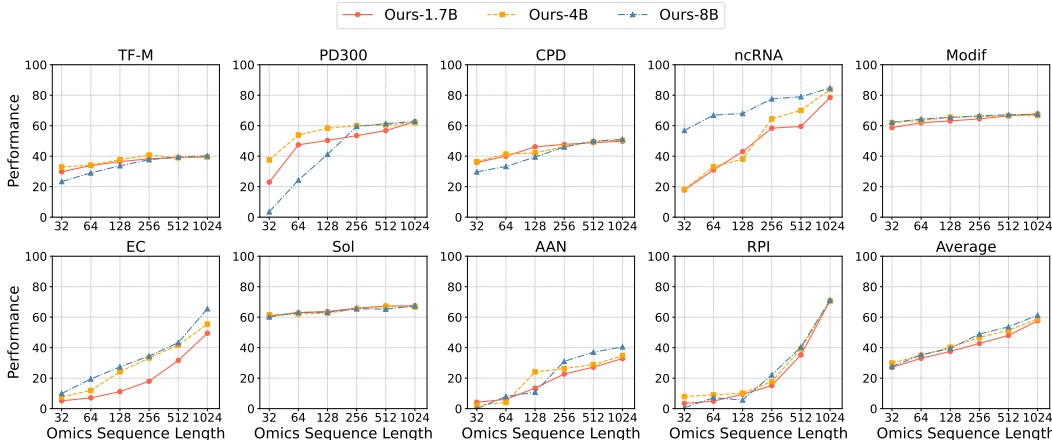

Figure 5: Performance of MOE across six maximum omics sequence-length settings. Results are reported for three backbone scales and nine tasks. Across nearly all tasks, performance improves as the readable sequence length increases, with larger gains for tasks sensitive to long-range or cross-sequence signals (e.g., ncRNA, EC, AAN, RPI). Several relatively local tasks (e.g., PD300, Sol) show earlier saturation, indicating diminishing returns at very long lengths.

significantly once a certain amount of data is reached. For example, for EC, Fmax is 4.02 and 5.98 at 20% and 30% of the data, respectively, but rises substantially to 24.91 and 40.92 at 40% and 50%, showing a sharp increase in performance once sufficient data coverage and diversity are provided. This highlights the high sensitivity of such tasks to data quantity and diversity, and emphasizes that certain tasks only show noticeable improvements once a minimum data scale is reached.

**Omics Sequence Length.** To translate the image-resolution effect from VLMs to MOE, we test whether the maximum readable length of biological sequences affects performance. We vary the maximum sequence length over six settings and evaluate three backbone scales (Fig. 5). Overall, performance increases as the model processes longer and more complete sequences, indicating that richer sequence context improves accuracy and robustness.

The gains are task dependent. Tasks that rely on dispersed regulatory signals or long-range dependencies (e.g., ncRNA and EC) benefit markedly from longer inputs, showing steady or even accelerated improvements at larger lengths. Interaction-oriented tasks (AAN, RPI) also improve as the model can jointly ingest longer partnered sequences, with notable jumps once the context exceeds mid-range lengths, suggesting threshold effects for capturing cross-sequence cues. In contrast, several relatively local or lower-complexity tasks (e.g., PD300, Sol, and CPD) tend to saturate earlier; beyond a moderate length, additional tokens yield diminishing returns. These patterns are consistent across different backbones, with larger models extracting slightly more benefit from extended context, presumably due to greater capacity to model long-range structure.

# 5 CONCLUSION

In this paper we systematically review prior work and categorize MOLLMs by omics encoding into NOE, SOE, and MOE, and benchmark their ability to understand multi-omics sequences across diverse tasks. Empirically MOE delivers the most robust performance among existing general LLMs and MOLLMs, and narrows the performance gap against omics-specific models. Extensive experiments further show MOE is a promising research direction for designing MOLLMs that can outperform specialist biological models in the future, which will be a landmark achievement in generalist-specialist learning paradigm. In the following work, we will further consider incorporate chain-of-thought technique into the MOLLM training.

## 6 REPRODUCIBILITY STATEMENT

All experiments are conducted on the Biology-Instructions corpus (He et al., 2025). To ensure reproducibility, our code is open-sourced and available at https://anonymous.4open.science/r/biomllm.

## 7 ETHICS STATEMENT

This study exclusively uses non-sensitive data that contain no human identifiable information. The study is limited to in-silico analyses of anonymized molecular sequences; no human or animal subjects were recruited, and no clinical data were accessed. The resulting models are intended for research purposes only and are not meant to guide medical decisions or any form of genetic intervention. We declare no competing financial or personal interests, and every experiment was carried out in accordance with current AI-research ethical standards.

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

## A  USAGE OF LLM

We used a large language model only for language polishing and improving readability (e.g., grammar, wording, and title phrasing).

## B  DATASET DETAILS

### B.1  TASKS DEFINITION

#### B.1.1  DNA TASKS

**Promoter Detection 300** This task involves detecting promoter regions within a 300 base pair (bp) window, which includes both the core promoter region and the surrounding regulatory elements.

**Core Promoter Detection** This task focuses on detecting a shorter, core sequence (usually around 50-100 bp) directly upstream of the transcription start site. Both tasks are important for understanding gene regulation and can aid in studying transcriptional activity, identifying novel genes, and mapping gene expression patterns. Model performance evaluation utilizes MCC, which captures the model's ability to predict promoter existence across different sequence contexts in a balanced manner.

**Transcription factor binding site detection in mouse** This binary classification task determines whether specific regions contain transcription factor binding sites in DNA sequences. These transcription binding sites (TBS) are critical for controlling the initiation, enhancement, or repression of transcription. The dataset from DNABERT-2 is utilized for this task Zhou et al. (2024), containing numerous DNA sequences with annotated TBS information. Model performance is evaluated using MCC, providing a fair measurement of the model's ability to discover TBS in diverse DNA sequences.

#### B.1.2  RNA TASKS

**Non-coding RNA Function Classification** This multi-label classification task predicts the functional class of non-coding RNA (ncRNA) sequences. The model outputs one or more class labels from a set of 13 possible ncRNA classes, including 'tRNA', 'miRNA', and 'riboswitch'. Accurate ncRNA classification is essential for understanding their regulatory roles in gene expression and their contributions to diverse biological processes and diseases. The nRC (non-coding RNA Classifier) dataset from (Fiannaca et al., 2017) is adopted for this task, utilizing features derived from ncRNA secondary structures. The output assigns each RNA sequence to one or more functional classes, enabling detailed examination of functional diversity within ncRNAs. Model performance is evaluated using accuracy (Acc), reflecting the classification capability across all categories.

**Modification Prediction** This multi-label classification task predicts post-transcriptional RNA modifications from RNA sequences. The model outputs one or more modification types from a set of 12 widely occurring RNA modifications, including 'm6A', 'm1A', and 'm5C'. Precise identification of RNA modification sites is essential for understanding RNA regulatory mechanisms and their roles in biological processes. The MultiRM dataset from (Song et al., 2021a) is employed, containing RNA sequences annotated with multiple modification types. Model performance is evaluated using the Area Under the Curve (AUC), capturing the prediction capability across different modification contexts.

#### B.1.3  PROTEIN TASKS

**Enzyme Commission Number Prediction** This multi-label classification task predicts enzyme functions by annotating protein sequences with corresponding EC numbers. The DeepFRI's (Gligorijević et al., 2021) EC annotation dataset from PDB chains is adopted, where binary multi-hot vectors are converted into corresponding EC numbers for language model compatibility. Performance evaluation utilizes the Fmax metric. Accurate EC number prediction is crucial for understanding enzyme catalytic functions and accelerating the discovery of novel enzymatic activities, with applications in biotechnology, industrial enzyme optimization, and drug development.

**Stability Prediction** This regression task assesses the intrinsic stability of proteins under various conditions, with each protein sequence mapped to a continuous stability score reflecting how well the protein maintains its fold above a certain concentration threshold (e.g., EC50). The dataset from Rocklin et al. (Rocklin et al., 2017) is employed, containing protease EC50 values derived from experimental data. Model performance is assessed using Spearman's correlation. Protein stability prediction is essential in protein engineering, particularly for therapeutic applications where protein integrity is crucial, reducing the need for experimental screening and facilitating the design of stable proteins for various applications.

### B.1.4 MULTI-OMICS TASKS

**RNA-Protein Interaction Prediction** This binary classification task identifies interactions between non-coding RNAs (ncRNAs) and proteins based on their sequences. Since most ncRNAs interact with proteins to perform biological functions, inferring these interactions facilitates understanding of biological activities involving ncRNAs (Li et al., 2016). The dataset from (Han & Zhang, 2023a) is used, comprising 14,994 samples. The evaluation metric employed is MCC.

**Antibody-Antigen Interaction Prediction** This binary classification task determines interaction relationships between antibodies and antigens based on their sequences. The objective is to establish correspondence between antigens and antibodies and predict effective antibody characteristics for new viral variants. The dataset sourced from (Zhang et al., 2022a) contains 22,359 antibody-antigen pairs. MCC is employed for performance assessment.

### B.2 DATASET DIVISION

The datasets for each task were partitioned into training, validation, and test sets following established benchmarks to ensure comprehensive evaluation of model performance across diverse biological sequences. Tab. 3 provides a detailed breakdown of the dataset sizes for all tasks.

For DNA-related tasks, the datasets consist of substantial collections with approximately 100,000 sequences for promoter detection tasks (PD300 and CPD), while the transcription factor binding site detection in mouse (TF-M) contains around 80,000 training samples. All DNA tasks maintain consistent validation and test set sizes of approximately 10,000-12,000 samples each.

RNA tasks exhibit diverse dataset scales, with Modification Prediction (Modif) featuring the largest training set of over 300,000 sequences, reflecting the abundance of available RNA modification data. In contrast, Non-coding RNA Function Classification (ncRNA) utilizes a more focused dataset of approximately 5,670 training samples but includes a larger test set of 4,840 samples for thorough evaluation.

Protein tasks include Solubility Prediction (Sol) with approximately 62,000 training sequences and Enzyme Commission Number Prediction (EC) with around 15,500 training samples. The multi-omics tasks, Antibody-Antigen Neutralization (AAN) and RNA-Protein Interaction Prediction (RPI), contain 22,359 and 14,994 training samples respectively, reflecting the specialized nature of these interaction prediction tasks.

In total, the benchmark encompasses approximately 695,000 training samples, 49,500 validation samples, and 51,100 test samples across all tasks, providing a robust foundation for evaluating model generalization capabilities across different biological domains and task types.

### B.3 TASK TYPES AND EVALUATION METRICS

### B.3.1 BINARY CLASSIFICATION

This type of task asks the model to predict one of two possible classes. In our case, either positive or negative. The evaluation pipeline involves first classifying via keywords based on the presence of predefined positive or negative keywords. If keywords classification fails, the pre-trained sentiment analysis model will be utilized as a fallback to determine the class based on the sentiment polarity assigned with a higher probability score.

| Task | Training/Validation/Test |
|------|--------------------------|
| **DNA Tasks** | |
| Promoter Detection 300 (PD300) | 94,712/11,840/11,840 |
| Core Promoter Detection (CPD) | 94,712/11,840/11,840 |
| Transcription factor binding site detection in mouse (TF-M) | 80,018/10,005/10,005 |
| **RNA Tasks** | |
| Modification Prediction (Modif) | 304,661/3,599/1,200 |
| Non-coding RNA Function Classification (ncRNA) | 5,670/650/4,840 |
| **Protein Tasks** | |
| Solubility Prediction (Sol) | 62,478/6,942/2,001 |
| Enzyme Commission Number Prediction (EC) | 15,551/1,729/1,919 |
| **Multi-Omics Tasks** | |
| Antibody-Antigen Neutralization (AAN) | 22,359/1,242/3,301 |
| RNA-Protein Interaction Prediction (RPI) | 14,994/1,666/4,164 |
| **Total** | |
| All | 695,155/49,513/51,110 |

Table 3: Data size (training/validation/testing) for each task in the benchmark. The table lists the four tasks by category: DNA, RNA, Protein, and Multi-Omics; the number of samples is given after each task in the format of "training/validation/testing". The "All" row sums the sample size for all tasks. The values in the table are raw sample counts, which are used in our paper to systematically evaluate model performance across different tasks and data sizes.

- **Matthews Correlation Coefficient (MCC):** Provides a balanced measure for binary classifications, even when classes are imbalanced. The metric ranges from -1 to 1, where -1 indicates perfect inverse correlation, 0 indicates random predictions or no correlation, and 1 indicates perfect positive correlation.

  Given true positives $TP$, true negatives $TN$, false positives $FP$, and false negatives $FN$, the MCC is computed as:

  $$MCC = \frac{TP \times TN - FP \times FN}{\sqrt{(TP + FP)(TP + FN)(TN + FP)(TN + FN)}} \tag{2}$$

- **Accuracy Score (ACC):** Calculates the proportion of correct predictions out of all predictions made. It ranges from 0 to 1, where 0 indicates no correct predictions, 1 indicates all correct predictions, and 0.5 as random predictions.

  Given the total number of correct predictions $C$ and total predictions $N$, the accuracy is computed as:

  $$ACC = \frac{C}{N} \tag{3}$$

### B.3.2 Multi-class Classification

This type of task asks the model to assign each input to one of several classes. In the non-coding RNA family prediction task, the model is required to predict one of 13 classes.

- **Accuracy Score (ACC):** Calculates the proportion of correct predictions out of all predictions made. It ranges from 0 to 1, where 0 indicates no correct predictions, 1 indicates all correct predictions, and 0.5 as random predictions.

  For multi-class classification with $K$ classes, let $C_k$ represent the number of correct predictions for class $k$ and $N$ be the total number of predictions. The accuracy is computed as:

  $$ACC = \frac{\sum_{k=1}^{K} C_k}{N} \tag{4}$$

### B.3.3 MULTI-LABEL CLASSIFICATION

This type of task involves inputs that may belong to multiple classes and asks the model to predict all of them. The evaluation process includes first extracting all relevant labels from the model outputs and converting them into binary multi-hot vectors representing the presence or absence of each class.

- **Area Under the ROC Curve (AUC):** Measures the model's ability to distinguish between classes across all thresholds. The metrics range from 0 to 1, where 1 indicates perfect ability to distinguish classes and 0.5 as random performance.

  Let $TPR(t)$ denote the true positive rate and $FPR(t)$ denote the false positive rate at threshold $t$. The AUC is computed as:

$$AUC = \int_0^1 TPR(FPR)dFPR \tag{5}$$

- **Fmax Score (Fmax):** Represents the maximum F1 score over all possible thresholds, providing a balanced measure of precision and recall in multi-label settings. The metric ranges from 0 to 1, where 0 indicates the worst balance of no correct predictions and 1 indicates a perfect balance between precision and recall.

  Given precision $Precision(t)$ and recall $Recall(t)$ at threshold $t$, the Fmax score is computed as:

$$Fmax = \max_t \left\{ \frac{2 \cdot Precision(t) \cdot Recall(t)}{Precision(t) + Recall(t)} \right\} \tag{6}$$

## C ADDITIONAL EXPERIMENTAL SETTINGS AND DETAILS

### C.1 TRAINING CONFIGURATION

| Component | Value |
|---|---|
| Max length | 3072 tokens |
| Omics window | 1024 tokens |
| Batch/GPU | 2 (train), 4 (eval) |
| Grad accum | 4 steps |
| LR | 3e-5, 10% warmup |
| Epochs | 5 |
| Precision | bf16 + Flash-Attention-2 |
| DeepSpeed | ZeRO-2 |

Table 4: Key Hyper-parameters and Training Configuration.

### C.2 DETAILED DESCRIPTION OF BASELINES

We provide detailed descriptions of all baseline models used in our experiments, organized into three categories: (1) omics-specific models, (2) general large language models (LLMs), and (3) multi-omics large language models (MOLLMs), further divided into non-omics-encoding (NOE), single-omics-encoding (SOE), and multi-omics-encoding (MOE) types.

**Omics-Specific Models** These models are domain-specialist architectures designed for specific biological tasks. They do not rely on natural language pretraining and are typically trained from scratch or fine-tuned on task-specific data.

- **DNABERT2** (Zhou et al., 2023): DNABERT-2 is a foundational language model pretrained on an extensive collection of genomic sequences spanning 2,063 species. It incorporates key innovations such as byte-pair encoding (BPE) for tokenization and Attention with Linear Biases (ALiBi) for position awareness, which together enhance computational efficiency. The model delivers state-of-the-art, ready-to-use representations for a range of genomic prediction tasks, including the identification of promoters, splice sites, and enhancers.

- **GCN** (Rossi et al., 2019): The GCN model addresses the challenge of classifying non-coding RNA function by leveraging their structural information. It represents each RNA molecule as a graph, where nodes correspond to structural elements and edges capture their complex interactions. A two-layer Graph Convolutional Network then processes this graph to generate a functional prediction, achieving state-of-the-art accuracy on the Rfam benchmark and identifying the key structural motifs that contribute to the classification.

- **MultiRM** (Song et al., 2021b): MultiRM introduces a multi-label deep learning framework for the simultaneous prediction of numerous RNA modification sites. Its architecture is centered on a shared bidirectional GRU encoder that learns a common representation of the RNA sequence, followed by an attention layer that specializes this information for each modification type. This design eliminates the need for separate, single-task models, providing a computationally efficient solution that uncovers both shared and modification-specific sequence determinants across species.

- **SaProt-GearNet** (Su et al., 2024): A structure-aware protein transformer that integrates sequence and structural information for enzyme function prediction. The hybrid transformer-GNN pipeline first enriches raw sequences with 3-D structural contexts via SaProt's vocabulary, then refines these embeddings through GearNet's message-passing layers to capture both local motifs and global topology.

- **DeepSol** (Khurana et al., 2018): DeepSol is an end-to-end deep learning framework that predicts protein solubility directly from primary sequence data. By employing a convolutional neural network (CNN) architecture, it automatically learns discriminative features from amino acid sequences, achieving superior predictive accuracy over traditional methods and facilitating the identification of proteins with high expression yields.

- **DeepAAI** (Zhang et al., 2022b): DeepAAI is a deep learning framework that predicts antibody-antigen interactions by learning adaptive relational graphs. The model constructs separate graphs for antibodies and antigens based on their sequences, and then employs Laplacian smoothing to propagate information, allowing it to infer functional representations for unseen antibodies. This graph-based methodology facilitates accurate prediction of neutralization activity and binding affinity.

- **ncRPI-LGAT** (Han & Zhang, 2023b): The ncRPI-LGAT model predicts ncRNA-protein interactions through a multi-step graph representation learning process. It begins by constructing a local network for each target RNA-protein pair and then transforms this subgraph into its corresponding line graph, where edges become nodes. A lightweight graph attention network then operates on this line graph to learn embeddings for each RNA-protein pair node by aggregating information from its neighbors.

**General Large Language Models**    These are general-purpose language models without domain-specific biological pretraining. They are evaluated in a zero-shot or few-shot setting on biological tasks.

- **DeepSeek-R1** (DeepSeek-AI, 2025): This model enhances its reasoning prowess through a sophisticated training regimen that includes multi-stage preparation and cold-start data integration prior to large-scale reinforcement learning. This approach refines the model's problem-solving strategies and output coherence, enabling it to excel in domains requiring rigorous logical inference and precise computation.

- **GLM4-9B-Chat** (GLM et al., 2024): GLM4-9B-Chat is a 9-billion-parameter large language model specifically fine-tuned for conversational interactions. It is designed to provide high-quality, responsive dialogue in both Chinese and English, serving as a versatile foundation for bilingual chat applications and assistants.

- **GPT-4o** (Hurst et al., 2024): Designed as a comprehensive multimodal AI, GPT-4o demonstrates state-of-the-art capabilities in understanding and generating content across text, code, images, and audio. Its performance extends to a wide range of languages and tasks. The model's development incorporates rigorous safety evaluations and alignment measures, ensuring its responsible deployment for a variety of applications.

- **Qwen3 Series** (Yang et al., 2025a): The Qwen3 Series represents a collection of open-source large language models, available in multiple parameter scales including 1.7B, 4B,

8B, and 32B. These models are trained on extensive datasets encompassing multiple languages and specialized scientific corpora, making them versatile foundation models for a wide range of applications.

**Multi-Omics Large Language Models**  These models integrate biological sequences with natural language processing capabilities. We categorize them based on their use of omics encoders.

NON-OMICS-ENCODING (NOE)  These models treat omics sequences as plain text and do not use biological encoders.

- **ChatMultiOmics** (He et al., 2025): ChatMultiOmics is an instruction-tuned large language model that processes DNA, RNA, and protein sequences as a unified text modality. By fine-tuning on a specialized corpus of biological instructions, it learns to interpret and reason about multi-omics data within a natural language framework, enabling it to perform a wide range of sequence analysis tasks through conversational interaction.

- **Intern-S1** (Bai et al., 2025a): Intern-S1 is a scientific large language model trained on interdisciplinary data spanning biology, chemistry, and physics. Uniquely, it processes complex scientific information directly without relying on domain-specific encoders for omics or other specialized data types, demonstrating strong generalization capabilities across multiple scientific fields.

- **NatureLM** (Xia et al., 2025): NatureLM is a large-scale scientific language model that employs a unified tokenization strategy to pretrain on DNA, RNA, and protein sequences concurrently. By processing these diverse biological modalities through a shared architectural framework without separate omics encoders, it establishes a cohesive representation of biological language, enabling cross-domain inference and generation capabilities.

SINGLE-OMICS-ENCODING (SOE)  These models use a single biological encoder (e.g., DNA or protein) aligned with an LLM.

- **BioReason** (Fallahpour et al., 2025): BioReason is a hybrid AI framework that combines a specialized DNA sequence encoder (NT-v2) with a large language model (LLM) to perform complex, multi-step biological reasoning. This integration enables the LLM to directly leverage structured genomic information, facilitating logical deduction and inference across diverse biological questions based on DNA sequence input.

- **ChatNT** (de Almeida et al., 2025): ChatNT introduces a unified biological sequence modeling approach by mapping RNA and protein sequences into DNA-like representations through a specialized DNA-centric encoder. This representation alignment enables seamless cross-omics reasoning within a coherent semantic space, allowing the model to integrate information across different biological modalities for comprehensive analysis.

- **ProLLaMA** (Lv et al., 2025):ProLLaMA is a protein-specialized language model that bridges protein sequence representations with natural language reasoning by aligning ESM2 embeddings into a Llama-based LLM. This alignment creates a unified model capable of interpreting protein sequences through the semantic understanding of a general-purpose language model.

C.3  DETAILED DESCRIPTION OF MAIN RESULTS

| Model | Task (Metric) | | | | | | | | |
|---|---|---|---|---|---|---|---|---|---|
| | **TF-M** | **PD300** | **CPD** | **ncRNA** | **Modif** | **EC** | **Sol** | **AAN** | **RPI** |
| Qwen3-1.7B | $\underline{39.55}^{\pm0.36}$ | $\underline{62.69}^{\pm0.94}$ | $49.80^{\pm0.31}$ | $78.50^{\pm0.22}$ | $\mathbf{67.85}^{\pm0.80}$ | $49.52^{\pm0.72}$ | $\mathbf{67.50}^{\pm0.78}$ | $32.85^{\pm2.25}$ | $70.70^{\pm0.54}$ |
| Qwen3-4B | $39.49^{\pm0.72}$ | $62.03^{\pm0.89}$ | $\underline{50.56}^{\pm0.44}$ | $\underline{83.75}^{\pm0.15}$ | $66.68^{\pm0.88}$ | $\underline{55.45}^{\pm0.41}$ | $66.69^{\pm0.71}$ | $\underline{34.76}^{\pm2.18}$ | $70.70^{\pm0.46}$ |
| Qwen3-8B | $\mathbf{40.20}^{\pm0.65}$ | $\mathbf{62.85}^{\pm0.46}$ | $\mathbf{50.97}^{\pm0.59}$ | $\mathbf{84.74}^{\pm0.06}$ | $\underline{67.51}^{\pm0.50}$ | $\mathbf{66.51}^{\pm0.53}$ | $\underline{67.39}^{\pm0.58}$ | $\mathbf{40.52}^{\pm2.10}$ | $\underline{71.20}^{\pm0.59}$ |

Table 5: Performance of Qwen3-Omics-SFT models on nine omics tasks (mean±std over three runs). Bold: best; underline: second-best.

## C.4 DETAILED DESCRIPTION OF DATA SCALING LAW EXPERIMENTS

| Data Scale | TF-M | PD300 | CPD | NcRNA | Modif | EC | Sol | AAN | RPI |
|---|---|---|---|---|---|---|---|---|---|
| **10%** | 30 466 | 6 247 | 8 001 | 9 471 | 9 471 | 2 235 | 1 555 | 1 499 | 561 |
| **20%** | 60 932 | 12 495 | 16 003 | 18 942 | 18 942 | 4 471 | 3 110 | 2 998 | 1 123 |
| **30%** | 91 398 | 18 743 | 24 005 | 28 413 | 28 413 | 6 707 | 4 665 | 4 498 | 1 684 |
| **40%** | 121 864 | 24 991 | 32 007 | 37 884 | 37 884 | 8 943 | 6 220 | 5 997 | 2 246 |
| **50%** | 152 330 | 31 239 | 47 356 | 40 009 | 47 356 | 11 179 | 7 775 | 7 497 | 2 807 |

Table 6: Training-set size (number of samples) used for each task when uniformly sampling 10%, 20%, 30%, 40%, and 50% of the original training data.

# D EXAMPLE SCENARIOS

## D.1 TF-M

**Task of TF-M**

> **Question**：
> <dna>AAAAAACAAACAAATCAAAAATGGGCAAAAAAAAAAAACCTAGCAAATAAAAAA
> CCAAAAAACAACAAAACACAAAAAAAAGCAGCAAAAAAAAAAAAAAACAG</dna>: Assess
> this DNA sequence for the presence of transcription factor binding sites.
>
> **Ground Truth**：
> Nope, I've not spotted any transcription factor binding sites in there.

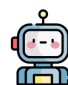

**Ours**

My assessment shows that this sequence does not include areas where transcription factors can bind.

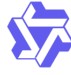

**Qwen3-32B**

CAGCAAAAAAAAAAAAAAACAG<dna>: Assess this DNA sequence for the presence of transcription factor binding sites.\nassistant\n<think>\n\n</think>\n\nAssessing a DNA sequence for **transcription factor binding sites (TFBSs)** involves identifying short, conserved DNA motifs known to interact with specific transcription factors. These motifs are typically 6–20 base pairs in length and are often **palindromic** or have **specific sequence preferences** (e.g., TATA-box, GC-box, CAAT-box).

...

✅ Summary of Findings\n\n| Feature | Present? | Notes |\n|--------|----------|-------|

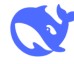

**DeepSeek-R1**

...

To get a definitive answer, you would need to:
Use Specialized Software: Run the sequence through a proper TFBS prediction tool like JASPAR, MEME Suite, or TRANSFAC. These tools compare your sequence against large databases of known motifs.
Experimental Validation: If this sequence is part of a suspected regulatory region, functional assays like EMSA (Electrophoretic Mobility Shift Assay) or ChIP (Chromatin Immunoprecipitation) would be required to confirm protein binding.
Genomic Context: The most important question is: Where is this sequence from? If it is upstream of a gene's transcription start site, it might be worth investigating experimentally. If it's from an intron or an intergenic region far from any gene, it is almost certainly non-functional for transcription factor binding.

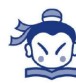

**Intern-S1**

To assess the provided DNA sequence for transcription factor binding sites (TFBS), we systematically analyze it for known consensus motifs and structural patterns associated with transcription factor recognition. Here's the breakdown:

...

**Final Answer:**
The DNA sequence contains potential transcription factor binding sites, including a **variant CCAAT box** (positions 51–55) for NF-Y and a **GC-rich motif** (positions 22–27) resembling an SP1 site. Additionally, extensive **poly-A tracts** may interact with AT-specific factors. However, these are partial matches, and functional validation is required.

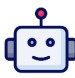

**NatureLM**

A a.

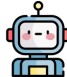

**ChatNT**

1.31

## D.2 PD300

### Task of PD300

**Question：**
<dna>CAGCCTCTCGACCGCCGGCCACCCGCCAGCCACGCGCGCCCAGACAGGAGGGCG
GCCGGGCGCTCGGCGGGGCGGGGCCTGCCGGGGGCGGGGCCTGCGGGCGGGGCGGGG
CCGCGGGAGGGCGGGGCCGGCGCGGCGAGCGCACCAGCAGCATCCTGGCTCAGCCGC
GGCGGTGGCGGGGGCGCAACCAGCGGGCCGAGGCGGCGCGCCAGCGGCGCCTTAAA
TAGCATCCAGAGCCGGCGCGGGGCAGGGAGTGGGCTGCAGTGACAGCCGGCGGCGGA
GCGGCCGGTCCACGGAGG</dna>: Is there any evidence of TATA box, Initiator, or other promoter-associated sequences here?

**Ground Truth：**
Promoter sequences are evident in the analyzed DNA sample.

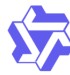

**Ours**

Examination reveals promoter elements in the DNA fragment.

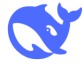

**Qwen3-32B**

assistant\n<think>\n\n</think>\n\nTo determine whether the given DNA sequence contains promoter-associated features such as the **TATA box**, **Initiator (Inr)**, or other regulatory elements, we need to carefully analyze the sequence for known consensus sequences and their positioning relative to the transcription start site (TSS).\n\n---\n\n###

...

**DeepSeek-R1**

Excellent question. This is a fundamental task in bioinformatics: scanning a DNA sequence for known regulatory motifs.
Let's analyze the sequence you provided.
Short Answer
No, there is no strong, canonical evidence of a TATA box or Initiator (Inr) element in this sequence. The sequence is overwhelmingly GC-rich and contains repetitive patterns that are more characteristic of a CpG island, which is often associated with promoters of "housekeeping" genes. These promoters typically lack TATA boxes.

...

Conclusion
The sequence CAGCCTCTCGACCGCCGGCC... does not contain evidence for a TATA box or a clear Initiator element. Instead, its defining features are:
Extremely high GC content.
Abundance of CpG dinucleotides, identifying it as a CpG island.
Multiple repeats of Sp1/KLF-like binding sites (e.g., GGCGG, GGGCGG).
This is a very strong signature of a TATA-less promoter commonly found for housekeeping genes. The transcription start site for such a promoter would likely be located within this GC-rich region, and initiation would be driven by the combined action of Sp1 and other factors binding to these GC-rich motifs, rather than by a TATA box/TFIID complex.

In summary: The evidence points away from a classic TATA-driven promoter and strongly towards a CpG island-associated, TATA-less promoter.

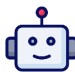

**Intern-S1**

Let's break down the analysis of the provided DNA sequence to determine if there are any promoter-associated sequences such as the TATA box or Initiator (Inr) elements.

...

### Final Answer
\[\boxed{\text{Yes, there is a TATA box (part of TTAAATAG), but no clear Initiator (Inr) sequence is found.}}\]

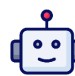

**NatureLM**

CCCAAG

**ChatNT**

The measured CPD nucleotide range for this sequence is 24.15.

## D.3  CPD

### Task of CPD

**Question:**
<dna>GGGAGAAAAACGGGGAACCAGGAATGTTGGGAGGGGGACGAGGAATACAAGGA
GGGAGGGGAACACTGAG</dna> Is this sequence located near the transcription start site?

**Ground Truth:**
Core promoter sequences are absent in the 70 bp DNA fragment.

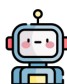

**Ours**

The 70 bp DNA sequence does not contain core promoter elements; characteristic motifs are absent.

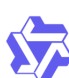

**Qwen3-32B**

user\n<dna>GGGAGAAAAACGGGGAACCAGGAATGTTGGGAGGGGGACGAGGAATACA
AGGAGGGAGGGGAACACTGAG<dna> Is this sequence located near the transcription start site?\nassistant\n<think>\n\n</think>\n\nTo determine if the DNA sequence \n\n`<dna>GGGAGAAAAACGGGGAACCAGGAATGTTGGGAGGGGGACGAGGAATACAA
GGAGGGAGGGGAACACTGAG<dna>` \n\nis located near a **transcription start site (TSS)**, we need to consider several key features commonly associated with promoter regions and TSSs, especially in **eukaryotic** or **prokaryotic** organisms (as it was not specified which one we're considering).\n\n---\n\n###

...

Conclusion\n\n- The sequence contains or is located **near a start codon (AUG)** when transcribed.\n- It is **GC rich**, which is often seen in promoter regions.\n- There is **no obvious TATA or CAAT box**, which could suggest a **TATA-less promoter**, common in some eukaryotic genes.\n- The presence of a **start codon** and the sequence structure **strongly suggest** that this DNA region is **located near the transcription start site (TSS)**, possibly in a **eukaryotic** gene.\n\n---\n\n### ✏️ Additional Steps for Confirmation\n\n1. **Align with known TSS annotation**: If this sequence is from a known genome, use tools like **Ensembl**, **UCSC Genome Browser**, or **CAGE** data to determine actual TSSs.\n2. **Check for RNA-Seq or

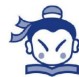

**DeepSeek-R1**

Excellent question. This is a classic example of a sequence that looks very much like it is located near a transcription start site.
The short answer is: Yes, it has very strong features that are almost always found in the promoter region, immediately upstream of the transcription start site.
Here's a breakdown of why, by analyzing the sequence features:

**...**

Conclusion
The sequence you provided exhibits the classic signature of a eukaryotic promoter, containing both TATA-box-like and multiple GC-box elements. Therefore, it is very likely located near the transcription start site, most likely directly upstream of it in the promoter region.
In summary: The evidence strongly points to YES.

**Intern-S1**

The solution I propose is Yes, it is.

**NatureLM**

CCCAAG

**ChatNT**

The measured CPD nucleotide range for this sequence is 24.15.

## D.4 NCRNA

### Task of ncRNA

**Question**：
<rna>GGATGCGATCATGTCTGCACTAACACACCGGATCCCATCAGAACTCCGAAGTTAA
GCGTGCTTGGGCGGGAGTAGTACTAGGATGGGCGACCCCTTAGGAAGTACTCGTGTTG
CATCCC<rna> Based on its structure, which RNA family does this sequence belong to?

**Ground Truth**：
The sequence is classified as belonging to the 5S_rRNA RNA family.

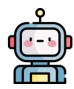

**Ours**

The sequence is identified as belonging to the 5S_rRNA class of non-coding RNA.

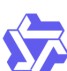

**Qwen3-32B**

assistant\n<think>\n\n</think>\n\nTo determine which **RNA family** the sequence you've provided belongs to, let's first analyze the **structure and features** of the given RNA sequence.\n\nSequence:\n```\nGGATGCGATCATGTCTGCACTAACACACCGGATCCCATCAG
AACTCCGAAGTTAAGCGTGCTTGGGCGGGAGTAGTACTAGGATGGGCGACCCCTTAGG
AAGTACTCGTGTTGCATCCC\n```\n\n---\n\n###

**...**

✅ Answer:\n\nThis RNA sequence likely belongs to the **Vault RNA** family.\n\n---\n\n### 💠
Vault RNA Overview:\n- Size: ~140–160 nucleotides\n- Structure: Conserved stem-loop motifs\n-
Function: Involved in drug resistance and intracellular transport in eukaryotes\n- Expression: Found
in mammals, especially in cancer cells\n\nIf you'd like to confirm, you can run a **BLASTN** or
**RNAfold** analysis to identify the specific RNA family or species.

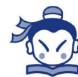

**DeepSeek-R1**

Let's break this down.
We have the RNA sequence:

**...**

**Final answer:**
\[
\boxed{4.5S\ SRP\ RNA}
\]

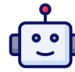

**Intern-S1**

I would respond by saying After analyzing the given sequence, the RNA family is identified as 7SK RNA.

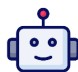

**NatureLM**

(null)

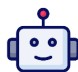

**ChatNT**

No, there is no evidence of a noncoding RNA family in this sequence.

## D.5  MODIF

### Task of Modif

> **Question**：
> <rna>GGAAAAAGTGACACCAATCCTGAGTCCTTTGGACACTGGATACTGGGCATCGTTC ATTGCCAGAAGTTTATAAAATCCTATTGGGAGAAAAGCAGGCTATC</rna> How does the sequence relate to predicted RNA modification types?
>
> **Ground Truth**：
> The predicted RNA modifications for include Cm.

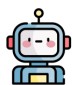

**Ours**

The RNA modifications predicted for the sequence are m5C.

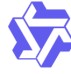

**Qwen3-32B**

types?\nassistant\n<think>\n\n</think>\n\nThe RNA sequence you provided:\n\n```\nGGAAAAAGTGACACCAATCCTGAGTCCTTTGGACACTGGATACTGGGC ATCGTTCATTGCCAGAAGTTTATAAAATCCTATTGGGAGAAAAGCAGGCTATC\n```\n\nis a 129-nucleotide RNA sequence. To determine how it relates to **predicted RNA modification types**, we must consider that RNA modifications

**...**

**Predicted Modification Types Summary**\n\n| **Modification** | **Potential Motifs / Context** | **Evidence in Sequence** | **Notes** |\n|----------------|----------------------------|------------------------|----------|\n| **m6A

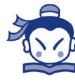

**DeepSeek-R1**

Excellent question. Analyzing an RNA sequence like this for potential modifications is a multi-step process that moves from general principles to specific, testable predictions.

Here's a detailed breakdown of how this sequence relates to predicted RNA modification types.

Summary Up Front

The sequence
GGAAAAAGTGACACCAATCCTGAGTCCTTTGGACACTGGATACTGGGCATCGTTCATTG
CCAGAAGTTTATAAAATCCTATTGGGAGAAAAGCAGGCTATC contains several sequence motifs that are known to be hotspots for common RNA modifications. The most prominent predictions would be for:

N6-methyladenosine ($m^6A$): High probability due to multiple DRACH motifs (especially GGACU).

A-to-I Editing: High probability due to a long stretch of adenosines (A-rich region) in a potentially double-stranded context.

Pseudouridylation ($\Psi$): Possible, depending on the secondary and tertiary structure of the RNA.

5-methylcytosine ($m^5C$): Less predictable from sequence alone, but potential sites exist.

It's crucial to remember that the final modification landscape is determined by the cellular context (which enzymes are expressed) and the RNA's secondary/tertiary structure.

...

Conclusion

The provided RNA sequence is rich in motifs associated with post-transcriptional modifications.

The strongest prediction is for N6-methyladenosine ($m^6A$) at specific DRACH sites (GGACA, GGACU, AAACU).

It also has a high potential for A-to-I editing in the TTTATAAAA region, contingent on the formation of a double-stranded structure.

Predictions for pseudouridylation and 5-methylcytosine are possible but require structural or experimental data for confirmation.

In a biological context, this RNA would likely be heavily modified, influencing its stability, splicing, translation, and function.

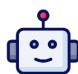

**Intern-S1**

I would conclude that The computational analysis of the sequence suggests the presence of the following RNA modifications: m1A.

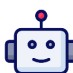

**NatureLM**

(null)

**ChatNT**

Yes, there is a clear and prominent presence of a modification in this sequence.

## D.6 EC

# Task of EC

**Question：**
<protein>MHHHHHHSSGVDLGTENLYFQSNAMDFPQQLEACVKQANQALSRFIAPLPFQNT
PVVETMQYGALLGGKRLRPFLVYATGHMFGVSTNTLDAPAAAVECIHAYSLIHDDLPAMD
DDDLRRGLPTCHVKFGEANAILAGDALQTLAFSILSDANMPEVSDRDRISMISELASASGIA
GMCGGQALDLDAEGKHVPLDALERIHRHKTGALIRAAVRLGALSAGDKGRRALPVLDKY
AESIGLAFQVQDDILDVVGDTATLGKRQGADQQLGKSTYPALLGLEQARKKARDLIDDAR
QALKQLAEQSLDTSALEALADYIIQRNK</protein> What enzymatic activity is associated with this protein sequence?

**Ground Truth：**
The enzyme's role in biochemical pathways is detailed by EC number EC2.5.1.-,EC2.5.1.10.

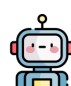

**Ours**

The types of reactions catalyzed by enzymes are classified under EC numbers. The type of reaction it catalyzes falls under EC number EC2.5.1.-,EC2.5.1.10.

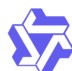

**Qwen3-32B**

user\n<protein>MHHHHHHSSGVDLGTENLYFQSNAMDFPQQLEACVKQANQALSRFIAPLP FQNTPVVETMQYGALLGGKRLRPFLVYATGHMFGVSTNTLDAPAAAVECIHAYSLIHDDL PAMDDDDLRRGLPTCHVKFGEANAILAGDALQTLAFSILSDANMPEVSDRDRISMISELAS ASGIAGMCGGQALDLDAEGKHVPLDALERIHRHKTGALIRAAVRLGALSAGDKGRRALPV LDKYAESIGLAFQVQDDILDVVGDTATLGKRQGADQQLGKSTYPALLGLEQARKKARDLI DDARQALKQLAEQSLDTSALEALADYIIQRNK<protein> What enzymatic activity is associated with this protein sequence?\nassistant\n<think>\n\n</think>\n\nTo determine the enzymatic activity associated with the given protein sequence, I will analyze it for conserved domains, motifs, and homology to known enzymes. Here's the step-by-step process and analysis:\n\n---\n\n###

...

Conclusion**\n\nBased on sequence analysis:\n\n> ✅ The protein sequence you provided is associated with **protein kinase activity**, most likely **serine/threonine-specific kinase**.\n\n---\n\n### 🔧 Further Recommendations\n\nIf you'd like to be more precise:\n- Perform a **BLAST search** using the NCBI database.\n- Use tools like **Pfam scan**, **SignalP**, **PhosphoSitePlus**, or **KinBase**.\n\nLet me know if you want to do that or if you're working on a specific protein name or organism.

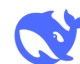

**DeepSeek-R1**

Excellent question. Analyzing a protein sequence to predict its function is a core task in bioinformatics. Based on the sequence you provided, here is a detailed breakdown of the analysis and the most likely enzymatic activity.
Summary Answer
The protein sequence you provided is highly likely to be a Serine Hydroxymethyltransferase (SHMT).
SHMT is a key enzyme in one-carbon metabolism that catalyzes the reversible conversion of the amino acid serine and tetrahydrofolate (THF) to glycine and 5,10-methylenetetrahydrofolate.

...

Conclusion
The presence of the definitive pyridoxal phosphate (PLP)-binding motif and the results of a BLAST search provide conclusive evidence that the protein sequence encodes Serine Hydroxymethyltransferase (SHMT), an enzyme central to one-carbon metabolism and amino acid interconversion.

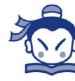

**Intern-S1**

As for my answer, it is Upon reviewing the provided protein sequence, the corresponding enzyme's activity is identified as <IUPAC> 1-deoxy-D-xylulose-5-phosphate synthase </IUPAC> activity, magnesium ion binding, thiamine pyrophosphate binding.

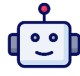

**NatureLM**

I -

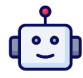

**ProLLaMA**

Superfamily=<Amidase signature (AS) superfamily>

1512
1513
1514
1515
1516
1517
1518
1519
1520
1521
1522
1523
1524
1525
1526
1527
1528
1529
1530
1531
1532
1533
1534
1535
1536
1537
1538
1539
1540
1541
1542
1543
1544
1545
1546
1547
1548
1549
1550
1551
1552
1553
1554
1555
1556
1557
1558
1559
1560
1561
1562
1563
1564
1565

### D.7  SOL

**Task of Sol**

**Question:**
<protein>MGSSHHHHHHSSGLVPRGSHMASMTGGQQMGRGSAMLPNTGRLAGCTVFITGA
SRGIGKAIALKAAKDGANIVIAAKTAQPHPKLLGTIYTAAEEIEAVGGKALPCIVDVRDEQQ
ISAAVEKAIKKFGGIDILVNNASAISLTNTLDTPTKRLDLMMNVNTRGTYLASKACIPYLKK
SKVAHILNISPPLNLNPVWFKQHCAYTIAKYGMSMYVLGMAEEFKGEIAVNALWPKTAIH
TAAMDMLGGPGIESQCRKVDIIADAAYSIFQKPKSFTGNFVIDENILKEEGIENFDVYAIKPG
HPLQPDFFLDEYPEAVSKKVESTGAVPELACGRTRAPPPPPLRSGC</protein> Solubility
prediction aims to assess if a sequence results in a soluble protein. Will this amino acid sequence
yield a soluble protein?

**Ground Truth:**
This protein has a high solubility potential.

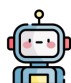

**Ours**

Solubility is an expected property of this protein.

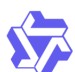

**Qwen3-32B**

user\n<protein>MGSSHHHHHHSSGLVPRGSHMASMTGGQQMGRGSAMLPNTGRLAGCTVF
ITGASRGIGKAIALKAAKDGANIVIAAKTAQPHPKLLGTIYTAAEEIEAVGGKALPCIVDVR
DEQQISAAVEKAIKKFGGIDILVNNASAISLTNTLDTPTKRLDLMMNVNTRGTYLASKACIP
YLKKSKVAHILNISPPLNLNPVWFKQHCAYTIAKYGMSMYVLGMAEEFKGEIAVNALWPK
TAIHTAAMDMLGGPGIESQCRKVDIIADAAYSIFQKPKSFTGNFVIDENILKEEGIENFDVYA
IKPGHPLQPDFFLDEYPEAVSKKVESTGAVPELACGRTRAPPPPPLRSGC<protein> Solubility
prediction aims to assess if a sequence results in a soluble protein. Will this amino acid sequence
yield a soluble protein?\nassistant\n<think>\n\n</think>\n\nTo determine whether a given amino
acid sequence yields a **soluble protein**, we can analyze it based on **sequence features
associated with solubility** and compare it to known solubility predictors. Here's how we can
assess the protein sequence you provided:\n\n---\n\n###

...

✅ **Final Prediction:**\n\n> **Yes**, this amino acid sequence is **likely to yield a soluble
protein**, especially under controlled expression conditions with proper tags and/or co-expression
with chaperones.\n\nLet me know if you'd like a detailed hydrophobicity plot or disorder prediction
for the sequence!

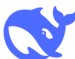

**DeepSeek-R1**

Based on the amino acid sequence you provided, here is a solubility prediction:
Prediction: This protein is likely to have LOW solubility.
Here is a breakdown of the reasoning:

...

Conclusion:
While the N-terminal His-tag will be useful for purification (likely under denaturing conditions if
the protein is insoluble), the intrinsic properties of the protein's amino acid sequence suggest a high
probability that it will form inclusion bodies (insoluble aggregates) when expressed in a system like
E. coli. To improve solubility, strategies like using a solubility-enhancing tag (e.g., MBP), lowering
the expression temperature, or trying different expression hosts would be highly recommended.

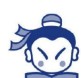

**Intern-S1**

I would say that The protein sequence you provided has been analyzed, and it is likely to be soluble.

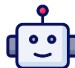

**NatureLM**

S

**ProLLaMA**

Superfamily=<NAD(P)-binding domain superfamily>

## D.8 AAN

### Task of AAN

Question：
Antibody-antigen binding assessment:
<protein>QVQLVQSGAEMKMPGASVKVSCKASGYTFTGNYIHWVRQAPGQGLEWMGWIA
PHSGDTSYAQRFQGRVTMTGDTSLSTAYMELSRLRSDDTAVYYCARGPFPNYYGPGSYW
GGLDFWGQGTLVSVSSXEIVMTQSPATLSVSLGERATLSCRTSQNVAYNFAWYQQKPGQA
PRLLIYEASSRATGTPARFSGSGFGTEFTLTISSMQSEDFAVYYCQQYNNWPSPFTFGPGTK
VHIKX</protein>> -
<protein>MRVMGILRSYQQWWIWGILGFWMLMICNVWGNLWVTVYYGVPVWKEAKTTL
FCASDAKAHKEEVHNIWATHACVPTDPNPQEIVLKNVTENFNMWKNDMVDQMHEDIISL
WDQSLKPCVKLTPLCVTLNCSDVKIKGTNATYNNATYNNNNTISDMKNCSFNTTTEITDK
KKKEYALFYKLDVVALDGKETNSTNSSEYRLINCNTSAVTQACPKVSFDPIPIHYCAPAGY
AILKCNNKTFNGTGPCNNVSTVQCTHGIKPVVSTQLLLNGSLAEEEVVIRFENLTNNAKIIIV
HLNESVEINCTRPSNNTRKSVRIGPGQTFFATGDIIGDIRQAHCNISRKKWNTTLQRVKEKL
KEKFPNKTIQFAPSSGGDLEITTHSFNCRGEFFYCYTSDLFNSTYMSNNTGGANITLQCRIKQ
IIRMWQGVGQAMYAPPIAGNITCKSNITGLLLTRDGGKEKNDTETFRPGGGDMRDNWRSE
LYKYKVVEIKPLGIAPDKAKRRVVEREKRAVGIGAVFLGFLGAAGSTMGAASMTLTVQAR
QLLSGIVQQQSNLLRAIEAQQHMLQLTVWGIKQLQTRVLAIERYLKDQQLLGIWGCSGKLI
CTTAVPWNASWSNKSYEEIWGNMTWMQWDREINNYTNTIYSLLEESQNQQEKNEKDLLA
LDSWESLWSWFNITNWLWYIRIFIIIVGGLIGLRIIFAVLSIVNRVRQGYSPLSFQTLTPSPREP
DRLGRIEEEGGEQDRARSVRLVNGFLALAWEDLRSLCLFSYHRLRDLILIAARAAALLGRSS
LWGLQKGWEALKYLGSLVQYWGLELKKSAISLFDAIAITVAEGTDRIINIVQRISRAFYNIP
RRIRQGFEATLQ</protein>>

Ground Truth：
Based on multiple analytical methods, I conclude there's no Ab-Ag interaction.

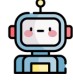

**Ours**

No, the antibody sequence does not exhibit features indicative of binding to this antigen.

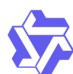

**Qwen3-32B**

VAEGTDRIINIVQRISRAFYNIPRRIRQGFEATLQ<protein>>\nassistant\n<think>\n\n</think>\n\
nTo assess **antibody-antigen binding**, we need to evaluate two sequences: the **antibody**
and the **antigen**. These are both long protein sequences in FASTA-like format.\n\nLet me parse
the sequences and perform a **binding assessment** based on known biological features

...

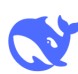

**DeepSeek-R1**

Of course. This is a detailed assessment of the potential binding between the provided antibody (a camelid VHH or "nanobody" sequence) and antigen (the HIV-1 gp41 protein).
Executive Summary
Based on the sequence analysis, the provided antibody fragment (a camelid VHH or potential nanobody) has a high probability of binding to the antigen (the HIV-1 gp41 envelope glycoprotein). The binding is likely specific and of high affinity, primarily targeting a key functional region of gp41 known as the membrane-proximal external region (MPER). This is a highly conserved and crucial area for viral fusion, making this a candidate for a broadly neutralizing antibody.

...

Conclusion
The antibody fragment described is highly likely to be a potent neutralizing agent against HIV-1 due to its structural characteristics (long CDR-H3) and a specific motif (YYGPGSY) that is a known signature for targeting the conserved and vulnerable MPER of the gp41 protein. This makes it a promising candidate for further therapeutic development.
Disclaimer: This is a in silico assessment based on sequence analysis and known antibody paradigms. Experimental validation (e.g., Surface Plasmon Resonance for affinity measurement, neutralization assays) is required to confirm binding and function. The anomalous X in the sequence should be clarified.

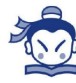

**Intern-S1**

The solution I propose is The binding assessment of the antibody and antigen is positive.

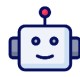

**NatureLM**

I

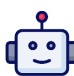

**ProLLaMA**

Superfamily=<Immunoglobulin-like domain superfamily>

## D.9   RPI

### Task of RPI

**Question：**
Can you predict if
<rna>AGTGCTATACTAGAGCTGGCTCCCAAGGACCCGCTGCCGCATTGCCTTTTGAAAC
AAAACAATGAACACGTTGGTAAAGGGGCCGTGCTTGTGTGTCGGTGACAAGGCGAGAT
CCCTGAGTCAGGTCAGGCTTGTAGATTCGAGTTCTGTTGCGAGTTTGATTGCCCCTCTG
ACTTTGTCCCCTGTACAACTAGGTTGATTAGGAATCAGCCAACTGTGTTCCCTGGGTGC
TCAGAAATCACAGCCCATATCCTCGAGAGGCCAAAATGAGAGCCAGGGGGGTTCCAAGA
TGAGTGGCTGCTTCTGGCCGGGAGCAGGTTTTCAAGTCATTAGAACACTCTGGCCTTTC
CTGGAGGTGATCTTGGAGCCATTCCTGCCCCTTTCAAGAGGAGTTAATGCCCAGCTCTG
TTTAGAGAAAATTGGGGGAGATGATTGCTCATGTGGGTGATAAGAATCACCTCCCGTG
CAGGGGTCTGCATAGAACACTCCATAGGCAAACCT</rna> will interact with
<protein>MALAGAPAGGPCAPALEALLGAGALRLLDSSQIVIISAAQDASAPPAPTGPAAPA
AGPCDPDLLLFATPQAPRPTPSAPRPALGRPPVKRRLDLETDHQYLAESSGPARGRGRHPG
KGVKSPGEKSRYETSLNLTTKRFLELLSHSADGVVDLNWAAEVLKVQKRRIYDITNVLEGI
QLIAKKSKNHIQWLGSHTTVGVGGRLEGLTQDLRQLQESEQQLDHLMNICTTQLRLLSEDT
DSQRLAYVTCQDLRSIADPAEQMVMVIKAPPETQLQAVDSSENFQISLKSKQGPIDVFLCPE
ETVGGISPGKTPSQEVTSEEENRATDSATIVSPPPSSPPSSLTTDPSQSLLSLEQEPLLSRMGSL
RAPVDEDRLSPLVAADSLLEHVREDFSGLLPEEFISLSPPHEALDYHFGLEEGEGIRDLFDCD
FGDLTPLDF</protein> based on their sequences and structures?

**Ground Truth：**
My answer is positive.

1674
1675
1676
1677
1678
1679
1680
1681
1682
1683
1684
1685
1686
1687
1688
1689
1690
1691
1692
1693
1694
1695
1696
1697
1698
1699
1700
1701
1702
1703
1704
1705
1706
1707
1708
1709
1710
1711
1712
1713
1714
1715
1716
1717
1718
1719
1720
1721
1722
1723
1724
1725
1726
1727

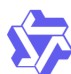

**Ours**

The interaction is predicted to be influenced by the presence of specific ions or molecules in the cellular environment.

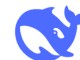

**Qwen3-32B**

TTDPSQSLLSLEQEPLLSRMGSLRAPVDEDRLSPLVAADSLLEHVREDFSGLLPEEFISLSPP HEALDYHFGLEEGEGIRDLFDCDFGDLTPLDF\<protein\> based on their sequences and structures?\nassistant\n\<think\>\n\n\</think\>\n\nTo determine whether the given RNA sequence will interact with the provided protein sequence, we need to consider a few key aspects

**...**

❗ Final Verdict:\n\n\> **Based on the sequences and structural motifs, it is plausible that the RNA and protein will interact.** However, more detailed computational or

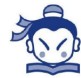

**DeepSeek-R1**

Unfortunately, based on the information you've provided (just the linear sequences), I cannot reliably predict whether this RNA and protein will interact.
Here's a detailed explanation of why this is a complex problem and what would be needed for a more accurate prediction.
The Short Answer
It is highly unlikely they interact in a specific, biologically meaningful way. The protein sequence (MALAGAPAGGPCAPALEALLG...) is MECP2, a crucial protein that binds to methylated CpG sites on DNA to regulate gene expression. The RNA sequence provided is a generic, seemingly random sequence without the specific structural or sequence motifs that MECP2 is known to recognize.

**...**

Conclusion
Based on the primary sequences alone and the known biology of MECP2:
Prediction: No specific, high-affinity interaction is expected.
Confidence: Low to medium, due to the lack of structural information. There could be weak, non-specific electrostatic interactions, but not a stable, functional complex.
To get a better prediction, you would need to provide the predicted or experimentally determined structures of both molecules.

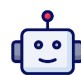

**Intern-S1**

As for my answer, it is Based on the analysis of the protein sequence, it seems that the protein is capable of binding RNA.

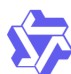

**NatureLM**

G C C A G C C A G G T G A G G G C C T G A G G G A G G C C C T G T G G T G A G G A A C T G A G G

