# OpenReview forum: "Rethinking Multi-Omics LLMs from the Perspective of Omics-Encoding"
_ICLR.cc/2026/Conference — Submitted to ICLR 2026_

### Official Review · Reviewer_ZU9g · 2025-10-15

**Soundness:** 3
**Presentation:** 2
**Contribution:** 2
**Rating:** 4
**Confidence:** 5

**Summary:**

This paper presents a comprehensive empirical study for Multi-Omics Large Language Models (MOLLMs), which are designed to understand biological sequences (DNA, RNA, protein) and natural language within a unified framework. The authors begin by systematically categorizing existing MOLLMs into three distinct paradigms based on their approach to encoding omics data: Non-Omics-Encoding (NOE), which treats sequences as plain text; Single-Omics-Encoding (SOE), which uses one shared biological encoder; and Multi-Omics-Encoding (MOE), which employs multiple, modality-specific encoders. A key contribution is the design and implementation of the first MOE model that integrates separate encoders for DNA, RNA, and protein. Through extensive benchmarking across nine diverse biological tasks, the paper rigorously quantifies the performance gap between generalist MOLLMs and specialized, task-specific omics models. The empirical results demonstrate that the proposed MOE architecture significantly outperforms other MOLLM variants and shows a promising potential to narrow this performance gap while maintaining the generality and natural language capabilities of a single, unified model. Further analysis provides valuable insights into cross-omics knowledge transfer, the benefits of multi-task training, and the scaling properties of MOLLMs.

**Strengths:**

This paper presents a systematic benchmark of Multi-Omics LLMs, introducing a novel MOE framework that demonstrates superior performance over existing paradigms by leveraging modality-specific encoders.

**1. Originality:** The work is original in its principled approach to integrating multi-omics data with LLMs. Instead of treating biological sequences as plain text, it proposes and validates the MOE paradigm, which uses dedicated encoders for DNA, RNA, and protein to project sequences into embeddings that are then aligned with the LLM. This demonstrates the feasibility and advantage of creating a unified representational space for natural language and diverse biological modalities.

**2. Quality:** The empirical evaluation of the proposed MOE framework is thorough and comprehensive. Beyond establishing its strong performance on a diverse set of nine tasks, the paper provides an in-depth analysis of its properties. This includes rigorous investigations into scaling laws (model size, data volume, sequence length), cross-omics knowledge transfer, and the benefits of multi-task training, offering valuable insights for future model design.

**3. Clarity:** The paper excels in its clear organization and presentation of the core concepts. The taxonomy of existing approaches (NOE, SOE, MOE) illustrated in Figure 1 is particularly effective, allowing readers to quickly grasp the landscape and the contribution of this work. The methodology and results are described with sufficient detail to ensure understanding.

**Weaknesses:**

While the paper makes a valuable contribution, several limitations hinder its impact and should be addressed to more fully achieve its stated goals.

**1. Limited Architectural Innovation:** The proposed MOE model, while effective, is architecturally simplistic. It employs a linear projection to align encoder outputs with the LLM, a method that has become standard in multimodal LLMs. The paper does not explore or justify this choice against more sophisticated alternatives (e.g., cross-attention layers, more complex adapters like Q-Former), nor does it provide an ablation study to show that this simple design is indeed optimal.

**2. Lack of Methodological Distinction in Training Paradigm:** The training methodology for the proposed MOE model bears a strong resemblance to that of existing NOE models like ChatMultiOmics, relying on the same instruction-tuning dataset (Biology-Instructions) and a similar supervised fine-tuning paradigm. The primary distinction lies in the input representation: replacing raw sequence tokens with embeddings from pre-trained specialist encoders. While this is a technically sound choice, it means the core language-centric training and reasoning process remains largely unchanged. The model does not introduce a novel pre-training or joint training strategy to foster a deeper, more intrinsic integration of omics knowledge into the LLM's reasoning fabric.

**3. Over-reliance on Imperfect Omics Encoders:** The paper's MOE paradigm is directly inspired by the success of Vision-Language Models (VLMs), which leverage powerful, general-purpose visual encoders that produce robust and semantically rich representations. However, this analogy overlooks a critical distinction: contemporary omics encoders (e.g., NT, ESM), while impressive, are not yet the "perfect“ feature extractors that their visual counterparts are. They operate on data that is inherently more sparse, noisy, and less understood, and their representations may be biased towards the specific pre-training tasks and datasets they were built on. Consequently, the performance of the entire MOE framework is critically contingent on the quality and biases of these external encoders. The paper does not address this dependency.

**4. Misplaced Terminology and Insufficient Model Ablation:** The paper incorrectly labels its scaling and sequence-length analyses as "ablation studies." A true ablation study would systematically remove or alter core components of the proposed MOE architecture to quantify their contribution. The current analyses, while valuable for understanding model properties, are scaling laws and sensitivity analyses, not ablations. The absence of a proper architectural ablation undermines the claim that the specific design of the MOE framework is rigorously validated, leaving its necessary components unclear.

**Questions:**

Based on the weaknesses and other concerns raised in the review, the following questions are posed to the authors for clarification and to strengthen the work.

**Main:**

1. The top-performing MOE model utilizes Qwen3-8B as its LLM backbone, while its primary competitor, ChatMultiOmics, is based on Llama-3.1-8B-Instruct. This difference in the core LLM introduces a significant confounding variable. Therefore, it is not clear whether the performance improvement is due to the changes in LLM. The authors also cannot prove through this set of experiments that using multi-omics embedding can achieve better performance than the sequence itself.

2. The training paradigm, which freezes the omics encoders and only updates the projection matrices and LLM, is adopted from LLaVA. This approach is justified in vision due to the maturity and generality of modern visual encoders. Given that omics encoders are a more recent and rapidly evolving technology, what was the rationale for strictly adhering to this frozen-encoder design? I think it's still worth discussing which components should be frozen and which ones shouldn't.

3. The paper effectively demonstrates that scaling the LLM component improves MOE performance. However, considering the relative novelty of omics encoders, it would be highly insightful to discuss the potential performance gains from similarly scaling or improving the encoders themselves. In-depth, it would be interesting to discuss whether the performance of the model is more affected by LLM or encoders.

4. The alignment between omics encoders and the LLM is achieved through a simple linear projection. While effective, this is a relatively simplistic fusion mechanism. Were more complex alignment architectures, such as employing cross-attention layers or modules akin to Q-Former, explored? If so, how did they compare to the linear projection? If not, could you discuss the reasoning behind focusing on this specific design choice?

5. A key question for any generalist model is its ability to generalize to unseen tasks. Was the MOE framework evaluated on any biological tasks not present in its instruction-tuning set? Demonstrating such task-level generalization would provide stronger evidence that the model is developing a fundamental understanding of multi-omics sequences, rather than merely memorizing patterns for the trained tasks.

6. Figure 4 indicates that the single-task performance on the AAN task is nearly zero. This raises a question about the multi-task learning: does including a task on which the model initially performs very poorly actually contribute positively to the learning process on other tasks, or could it potentially act as noise? Could you discuss the observed effect of the AAN task within the multi-task training mixture?

**Minor:**

1. Figure 3 panels (a) and (b) present two largely independent analyses (comparison with EVO2 and cross-omics transfer). Grouping them together in a single figure is somewhat confusing, and they might be better presented as separate figures for clarity.

2. The section titled "Ablation Study" (4.6) primarily investigates scaling laws and sensitivity, which differs from the conventional definition of an ablation study that involves removing core components to test their necessity. We suggest reconsidering the title of this section for greater precision.

---

### Official Review · Reviewer_RECn · 2025-10-31

**Soundness:** 2
**Presentation:** 2
**Contribution:** 3
**Rating:** 4
**Confidence:** 4

**Summary:**

This work introduces an LLM-based multi-omics encoding (MOE) model for DNA, RNA, and proteins, which specifically integrates biological encoders for each of these modalities with an LLM to support multi-omics understanding and natural language outputs. It benchmarks this model against general LLMs, omics-specific models, and LLM based single-omics multimodal models such as BioReason on various tasks. It is reported that the MOE model developed in this work outperforms general LLMs and single-omics multimodal models, however, the performance lags behind omics-specific models. The paper also conducts additional analyses to better understand the performance of their MOE model. For example, they conduct an ablation study which shows that performance improves with sequence length for nearly all tasks, and that performance generally also improves as the model’s size increases. They also conduct analyses which show that multi-task training improves performance on most tasks, and that training on multiple modalities can improve performance for DNA and RNA tasks, but not for protein-specific tasks.

**Strengths:**

•	The paper takes meaningful steps towards tackling an important and significant problem of bridging the gap between text and multi-omics, and leveraging synergies between different modalities.
•	This work has a significant degree of originality since enabling an LLM to work with multiple omics is an underexplored problem.
•	Ablation studies and additional analyses are well motivated and provide valuable insights into the strengths and limitations of multi-omics encoding, and scaling laws in this setting.
•	Good high level discussion of this area of research.
•	Figures and tables are generally clear and well designed.
•	Showing the example of how various models respond to questions in the supplement is very interesting.

**Weaknesses:**

•	Writing needs improvement. There are spelling and grammar errors throughout the paper to the point where it detracts from clarity and is distracting when reading. (There are too many to list here, but for instance it should be “model that can understand multi-omics data” on line 19, “Despite the above progress” on line 78, “We conduct experiments to facilitate research on MOLLMs” on line 93). Furthermore, the narrative of the paper is sometimes difficult to follow. For example, it isn’t clearly stated in the abstract that the authors actually develop and benchmark a novel MOE architecture.
•	How “MOLLMs” in Table 1 are benchmarked, and more specifically, how they are fine tuned is not clear, and it is therefore not convincing that the architecture proposed in this paper outperforms these models. For instance, BioReason is fine tuned using GPT-4o augmented training data for each downstream task in their original work (https://arxiv.org/pdf/2505.23579), and it is unclear if a similar setup is used in this work.
•	The results are interesting, but some are obvious. For example, biological signals that have known long range effects need longer inputs to capture that. It’s good that the models verify this common sense notion, but there could be more important relationships to focus main text space on.
•	The set of tasks selected is poorly motivated. Out of all the possible common tasks in biology, how were these tasks selected? How do we know that they weren’t selected in a way that biases the results? This could be addressed by including more tasks. How about protein function prediction tasks, like based on gene ontology? Or whether a protein has a transmembrane domain, or other protein domain? To truly address the questions posed in the paper, the field will need to have a much more comprehensive benchmarking suite of tasks.
Based on these weaknesses, I am providing an initial recommendation of rejection. However given the importance of the problem tackled and the originality of this work, it could be a valuable inclusion at ICLR if the authors can clarify and justify their approach to benchmarking other MOLLMs (see questions below), and if they can significantly improve the writing of this paper for the camera ready version.

**Questions:**

•	How did you fine tune the other MOLLMs in table 1 on the downstream tasks evaluated? If your strategy deviated from the original works these MOLLMs were presented in, can you justify why?
•	On a related note, works such as BioReason claim to outperform OSMs, however your work claims they do not. Can you explain if your benchmarking is a more fair evaluation of this than the original benchmarking? Are there any flaws you can identify with the original benchmarking?
•	Why was this particular set of tasks selected?

---

### Official Review · Reviewer_dXRY · 2025-10-31

**Soundness:** 3
**Presentation:** 2
**Contribution:** 2
**Rating:** 4
**Confidence:** 4

**Summary:**

This paper presents a comprehensive empirical study to determine the most effective architecture for Multi-Omics Large Language Models (MOLLMs), AI models that can understand natural language and biological sequences (like DNA, RNA, and protein).

**Strengths:**

- this paper provides a large-scale, systematic comparison of four different classes of models (General LLMs, NOE, SOE, MOE) against specialist "gold standard" models (OSMs). This is conducted across 9 distinct tasks, 3 modalities, and 3 different model sizes


-The paper goes beyond just claiming "MOE is best." It provides deeper insights, such as identifying "negative transfer" (where adding protein data hurt RNA task performance) and showing that multi-task learning, while mostly beneficial, can hurt specific tasks like EC (Enzyme Commission) prediction


-The paper proves that architecture is more important than just parameter count. The smallest 1.7B-parameter MOE model significantly outperformed a much larger 8B-parameter NOE model, demonstrating the high efficiency and effectiveness of the MOE design.

**Weaknesses:**

-it seems that the code files in the git repo are most empty at the time of reviewing

-Encoder Dependency: The MOE model's performance is fundamentally limited by the quality of its "frozen" encoders (NT-500M and ESM2-650M). If a better specialist encoder exists, the MOE model will be capped by its pre-trained knowledge.

-The paper uses a single linear projection matrix ($W_k$) to map the encoder embeddings into the LLM's embedding space. This is the simplest possible "alignment" method. It's parameter-efficient but may act as a representational bottleneck, struggling to effectively translate the rich, complex information from the biological encoders into a "language" the LLM can fully utilize. More powerful methods (like Q-Formers, perceiver resamplers, or multi-layer cross-attention) could yield much stronger fusion.

-would like to improve my score if these concerns are addressed

**Questions:**

- You used the NT-500M encoder for both DNA and RNA sequences. Given that RNA's function is heavily dependent on its secondary structure, which is not a primary feature of DNA, do you believe this choice created a performance bottleneck? Did you experiment with using a dedicated, structure-aware RNA foundation model?

-Your results show that training on all omics data (DRP) hurt performance on protein-only tasks (Figure 3b). Do you attribute this "negative transfer" to the simplicity of the linear projection, or is it possible that the NT-500M embeddings are introducing "noise" that confuses the LLM when it's trying to reason about the ESM2 protein embeddings?

-Your sequence length ablation (Figure 5) stops at 1024 tokens. This is quite short for many real-world genomic or protein sequences. Did you test longer contexts (e.g., 4k or 8k tokens), and does the performance scaling continue, or does the linear projection layer fail to handle such long-range dependencies effectively?

---

### Official Review · Reviewer_zgod · 2025-10-31

**Soundness:** 2
**Presentation:** 2
**Contribution:** 3
**Rating:** 2
**Confidence:** 4

**Summary:**

This paper presents a comprehensive empirical analysis of Multi-Omics Large Language Models (MOLLMs) and introduces a taxonomy of three encoding paradigms: Non-Omics Encoding (NOE), Single-Omics Encoding (SOE), and Multi-Omics Encoding (MOE). The authors benchmark their MOE prototype—integrating DNA, RNA, and protein encoders via projection into a shared LLM space—across nine omics and cross-omics tasks using the Biology-Instructions dataset (~700K samples). Results show that MOE consistently outperforms NOE and SOE, narrowing but not closing the gap with specialist models dataset.

**Strengths:**

The paper’s strengths lie in its comprehensive benchmarking, conceptual clarity, and rigorous scaling analyses(model size, data volume, and context length). The proposed taxonomy (NOE/SOE/MOE) helps organize the growing MOLLM landscape. Good experiment design and ablation study. Also the paper studies why cross-modality models, i.e., Single and Multi-Omics-Encoding, that integrate single/multi-omics and natural language, are worse than Omics-specific models that only use omics data without integrating any natural language, unlike other domains, such as vision-language models. This is a well-motivated and interesting topic to the community.

**Weaknesses:**

1-	Limited methodological novelty.
2-	Despite the motivation, no mechanistic or attribution-based analyses (e.g., motif-level, structure-level) are provided.
3-	Ambiguity in performance attribution: Gains could stem from model scale or task diversity rather than the multi-encoder design. Parameter-matched SOE baselines are missing
4-	Protein performance degradation is observed but not analyzed or mitigated (Negative transfer)
5-	The paper “rethinks” the space empirically but doesn’t introduce a fundamentally new modeling perspective.

Furthermore, the paper would be much improved by adding multi-omics-specific models, even the simplest one that concates the embeddings of DNA, RNA, and protein, without any natural language. By comparing multi-omics-specific to omics-specific models, we can observe the improvement from multiomics integration. By comparing multi-omics-specific and multi-omics-encoding, we can better isolate the effects of adding natural language. This would improve the analysis about Figure 3.

**Questions:**

1.	Can you provide interpretability evidence (e.g., attention or attribution maps) to support the claim of transparency/interpratability?
2.	How sensitive are results to encoder choice?
3. In Figure 5, why do smaller models perform better than large models when the length is short on PD300 but perform worse on ncRNA?

---

### Meta-Review · Area_Chair_1tLh · 2026-01-06

**Summary:**

While the problem is important, the proposed model lacks clear novelty and rigorous validation. The MOE architecture relies on a standard linear projection to connect omics encoders with the LLM, without justification or comparison to more expressive alternatives, and its training paradigm largely mirrors existing NOE-based models, differing mainly in input representation. Moreover, the approach is heavily dependent on imperfect pre-trained omics encoders, raising concerns that the LLM may be shielded from genuine biomedical understanding.

Finally, the paper mislabels scaling analyses as ablation studies and provides no true architectural ablation, making it difficult to assess the contribution of key components and to validate the reported results. Without a convincing rebuttal, these concerns remain insufficiently addressed.

**Reviewer Concerns:**

No rebuttal.

**Reviewer Scores:**

The novelty and the interpretation evaluation.

---

### Decision · Program_Chairs · 2026-01-26

Reject